# A quantitative, multimodal wearable bioelectronic device for comprehensive stress assessment and sub-classification

Xiaochang Pei[1,2,5], Anita Ghandehari[1,2,5], Shingirirai Chakoma[1,2,5], Jerome Rajendran [1,2,3,5], Jorge Alfonso Tavares-Negrete [2,4] & Rahim Esfandyarpour [1,2,3,4] ✉

Stress is a universal experience impacting mental and physical health. However, no precise, objective wearable tool exists for continuous, long-term stress monitoring, which is essential for understanding stress-related health outcomes. To address this gap, we introduce SQC-SAS, a multimodal wearable device that simultaneously and continuously measures multiple physiological and molecular stress biomarkers for quantitative stress assessment and sub-classification. This device features exceptional environmental stability, reusability, and fully wireless data and power operation. Machine learning enables data-driven stress assessment and classification across multiple stress states, allowing biomarker profiles to be correlated with each state. Its wristband-like design enables continuous stress monitoring and real-time visualization. We envision our wearable will greatly advance precise, objective stress assessment and monitoring, offering unprecedented capabilities and laying the foundation for personalized interventions and a deeper understanding of stress-related outcomes.

Stress is a common human experience that impacts people across all ages, professions, and cultures[1]. In 2022, 52% of Americans and 60% of individuals from 34 countries reported facing stress so overwhelming that they struggled to manage it at least once throughout the year[2]. Prolonged exposure to acute stress, caused by demanding physical or psychological events, can evolve into chronic stress, potentially leading to mental health disorders such as anxiety and depression, as well as physical conditions like cardiovascular disease and obesity[1,3,4]. Effective management of stress and its impact on individual health, the healthcare system, and society starts with accurate stress detection. However, there is currently no scientific measure or universally accepted quantitative definition of stress. Most existing clinical stress assessment methods, such as the perceived stress scale (PSS-10)[5] and immunoassay-based cortisol tests in body fluids[6], have notable limitations. Self-reported questionnaires are inherently subjective, qualitative, and prone to bias due to variations in individual perception, recall inaccuracies, and social desirability effects[7]. In contrast, physiological biomarkers such as cortisol, which reflect the human body's neuroendocrine response to stress, offer objective and quantifiable measures. However, conventional cortisol assessment methods, particularly blood tests, are invasive, require skilled medical personnel, specialized laboratory infrastructure, and provide limited insight as they represent only single-point snapshots[8]. Furthermore, both approaches lack the ability for continuous monitoring of dynamic physiological changes and are incapable of distinguishing between different stress subtypes.

In response to some of these limitations, recent research efforts have focused on the development of wearable sensor technologies for

[1]Department of Electrical Engineering and Computer Science, University of California, Irvine, CA, USA. [2]Laboratory for Integrated Nano Bio Electronics Innovation, The Henry Samueli School of Engineering, University of California, Irvine, CA, USA. [3]Henry Samueli School of Engineering, University of California, Irvine, CA, USA. [4]Department of Biomedical Engineering, University of California, Irvine, CA, USA. [5]These authors contributed equally: Xiaochang Pei, Anita Ghandehari, Shingirirai Chakoma, Jerome Rajendran. ✉e-mail: rahimes@uci.edu

stress monitoring to facilitate continuous, real-time assessments by leveraging various physiological signals. These wearable sensors typically measure parameters such as electrocardiograms (ECG)[9], galvanic skin response (GSR)[10], skin temperature (ST)[11], or radial artery pulse (RAP)[12], which hold the potential to provide continuous, dynamic, and objective indicators of stress. Despite these promising capabilities, a significant and common limitation shared by these wearable devices is their lack of specificity, resulting in false-positive and/or false-negative outcomes. For instance, physiological signals like ECG, RAP, and ST can also be influenced by confounding factors unrelated to stress, including physical activity, dietary habits, environmental conditions, or circadian rhythms, thereby influencing the accurate differentiation between stress-induced and non-stress-induced physiological changes[13–15]. With studies demonstrating strong correlations between cortisol levels in sweat and blood[8], sweat cortisol testing offers a powerful, convenient, and non-invasive alternative to blood testing without the specificity issues associated with stress-related biosignals. Recently, efforts have been made to develop cortisol-sensing devices employing detection methods such as colorimetric[16], fluorescence-based[17], antibody-based[18], and enzymatic[19] assays. However, these methods possess significant limitations, particularly their inability to provide continuous monitoring due to a lack of reusability and no applicability in wearable formats arising from environmental instability[20]. Biological components like antibodies or enzymes used in these assays exhibit poor environmental stability[21,22], degrading under temperature fluctuations, humidity variations, and pH changes common to wearable device usage outside controlled laboratory environments, thereby restricting their practical implementation. Recent research has explored the utilization of aptamers, which offer improved environmental stability compared to antibodies and enzymes[23–25]; nevertheless, aptamers present significant cost barriers. Furthermore, existing studies reliant on single-modal or isolated stress biomarker assessments, whether physiological or molecular, fail to capture the complex, nonlinear interplay and intricate temporal dynamics underpinning physiological and biochemical responses to stress, subsequently resulting in contradictory and inconclusive studies. Such reductionist approaches inadequately account for the nuanced regulatory mechanisms driven by critical neuroendocrine pathways, notably the hypothalamic-pituitary-adrenal (HPA) axis and the sympathetic-adrenal-medullary (SAM) axis[26]. Consequently, interpretations derived from these isolated biomarkers are compromised by limited specificity, high inaccuracy, and diminished clinical significance, severely restricting their utility in differentiating distinct stress states, such as acute versus chronic stress or psychological versus physiological stressors[27–29]. To address these fundamental limitations and substantially enhance their reliability and clinical applicability, studies must emphasize multimodal and introduce approaches that concurrently measure and analyze clinically relevant physiological and molecular stress biomarkers (MSB) together. Such integrative measurements are crucial for accurately capturing the full spectrum of stress-related responses, providing deeper insights, and enabling actionable stress assessments.

Some recent studies focused on wearable devices designed to monitor isolated molecular biomarkers related to stress (e.g., cortisol)[30,31], purely physiological signals (e.g., ECG, GSR)[10,32], or combinations of some physiological signals but with molecular markers irrelevant to stress responses (e.g., glucose)[33]. However, these studies still exhibit significant gaps, including the incapability to simultaneously measure stress molecular markers levels and physiological signals to provide comprehensive and accurate stress profiles. There is a lack of simultaneous multimodal monitoring of both relevant molecular and physiological stress biomarkers resulted in fragmented, incomplete, and often misleading stress evaluation studies[11,12,34]. Such gaps not only compromise the ability to quantitatively, objectively, and comprehensively define stress and assess stress levels but also

severely restrict the differentiation between specific stress subtypes, such as psychological from physiological stress[35,36]. Thus, there remains an urgent and critical need within the field to develop an innovative, integrated wearable capable of comprehensive (i.e., molecular and physiological signals), continuous, multimodal, and reusable sensing of biomarkers with environmental stability for prolonged use in a wearable format. Achieving such a sophisticated platform would enable a deeper understanding of stress physiology, define stress objectively, facilitate accurate identification and subclassification of stress states, and ultimately resolve persistent inconsistencies and contradictions prevalent in current stress research literature[37,38]. Such devices have yet to be realized. In addition, reported wearable stress-monitoring solutions encounter several other significant limitations that hinder their widespread adoption in practice. They commonly rely heavily on controlled laboratory conditions or require specific, activity-induced physiological stimuli (e.g., exercise)[23], thereby limiting their utility for continuous or spontaneous stress monitoring. A considerable proportion of them suffer from inadequate biocompatibility[39], raising potential risks for prolonged skin exposure, discomfort, and possible adverse reactions, thus impeding their feasibility for long-term continuous use. Additionally, their on-body battery-use nature further exacerbates these challenges due to reduced flexibility, frequent maintenance demands, short battery lifespans necessitating regular recharging or replacement, cumbersome form factors, and compromised user comfort[40].

To address some of the critical and long-standing challenges outlined, this study introduces a Smart Quantitative and Comprehensive Stress Assessor and Sub-Classifier Wearable (SQC-SAS). This flexible nanomaterials-based wearable bioelectronic device is capable of simultaneous and multimodal measurement of MSB alongside physiological stress indicators (PSI) biosignals continuously, automatically, long-term, and precisely. The SQC-SAS wearable also enables a quantitative and data-driven assessment of stress, improving the identification and differentiation of distinct stress subtypes. The SQC-SAS wearable's highly flexible microfluidic is designed for complete and contamination-free renewal of sweat samples in a few minutes (≤10 min), ensuring the accuracy and reliability of measurements. Additionally, this pioneering wearable solution stands out due to its wireless, wirelesslypowered, and autonomous operation, as well as flexible, nanomaterials-based, and biocompatible (>96% cell viability) construction that ensures maximum user comfort for reliable use. It also features a smart, power-efficient, and memory-equipped data collector and wireless reader (WLR), enabling continuous, and extensive storage of MSB and PSI stress biosignals for stress profile monitoring. Its real-time and on-demand wireless data transmission directly to smartphones also provides immediate visualization. The two features facilitate both instant assessments and retrospective analyses. Moreover, its ML analysis shows the potential of the collected MSB and PSI signals for quantitative, data-driven stress assessment and facilitating objective sub-classification of multiple stress states. Consequently, our work and introduced SQC-SAS wearable bioelectronic is expected to improve objective stress assessment, monitoring, and management, offering researchers a powerful tool for stress-related studies and laying a foundation for advanced, personalized healthcare interventions.

## Results

### Design and development of SQC-SAS wearable

The SQC-SAS wearable bioelectronic comprises three core components as shown in Fig. 1A: (1) The multimodal and multi-sensing (M&M) panel, which includes the PSI patch and MSB patch. This panel features biocompatible, skin-conformal, epidermal, miniaturized, flexible, and stretchable designs, equipped with multiple sensors for the simultaneous collection of multimodal PSI and MSB signals. (2) The battery-less on-skin data acquisition system (BLOSDA) that provides high data

throughput for multi-sensor readouts through wireless power and data transmission via near-field communication (NFC). This component acts as the data and energy bridge between the M&M panel and the WLR. (3) The battery-powered WLR, which manages wireless power and data exchange with the BLOSDA system, enables continuous, long-term, on-demand, real-time monitoring and storing data on a microSD card. The PSI sensor patch was constructed using mechanical patterning to build serpentine-shaped electrodes and interconnections on ultrathin (15 μm) gold-coated polyimide (Au/PI) films (Fig. 1B, C). One pair of serpentine electrodes functions as the GSR sensor, while another pair of back-to-back serpentine electrodes serves as the ECG sensor. The RAP sensor is made from a MXene/PANI/PEDOT-based e-textile integrated with interdigitated electrodes (IDE) composed of Au/PI film. The micron-range thickness and serpentine-shaped design of the PSI patch render it flexible, stretchable (up to 94% tensile strain), and epidermal, creating a reliable skin-electrode conformal interface that benefits signal acquisition and ensures the sensors on the PSI patch are comparable to commercial products. The MSB sensor patch contains the electrode arrays modified with molecularly imprinted polymer (MIP) layers, two IP electrodes, and a multi-inlet microfluidic system, as depicted in Fig. 1D. The MIP layers are synthesized through an electro-polymerization process that integrates functional monomers and redox probes, resulting in a precise and selective sensing mechanism. The inclusion of Prussian blue (PB) within the MIP matrix serves as an intrinsic electrochemical signaling probe, enabling rapid, low-voltage biomolecule sensing without the overoxidation issues common in traditional sensors. This approach enhances the reusability and reusability of the sensor. The IP electrodes are loaded with iontophoresis hydrogel (IP gel) to induce sweat autonomously anytime without the need for strenuous exercise. Carbachol gels (Carbagel) are preferred over conventional pilocarpine gels (Pilogel) due to their superior efficiency in sustaining prolonged sudomotor axon reflex sweat secretion, making them optimal for microfluidic sweat sampling. Our study demonstrated carbachol-induced sweat at a rate of 4 μl/min, alongside a high biocompatibility of 96% cell viability, validating its suitability for wearable applications (Supplementary Fig. 1). The mechanically engraved microfluidic system was designed to collect sweat samples induced by IP, preventing contamination and evaporation to enable on-site cortisol level detection.

The M&M panel was connected to the BLOSDA unit for wireless data acquisition and signal processing, eliminating the need for rigid or bulky wiring to improve the device's comfort and wearability. As illustrated in the functional block diagram (Fig. 1E), the BLOSDA integrated various key components: analog front-end (AFE) circuits for ECG and GSR signal amplification and filtering, a resistance-to-digital converter (R2D) chip for RAP signal monitoring, an electrochemical front end (FE) for cortisol level detection and sensor restoration, an on-board temperature sensor for body temperature monitoring, a booster with current source for IP, and microcontroller unit (MCU) for analog-to-digital conversion, data processing, communication, and power management. The NFC module on BLOSDA serves dual purposes: it harvests power from the WLR or smartphone to operate the BLOSDA unit, and it enables real-time, wireless high-rate data collection/transmission (up to 40 kbit/s) from multiple sensors on the M&M panel to smartphone or WLR. A custom Android app was developed to visualize monitored signals in real-time on a smartphone, while the WLR is designed to attach directly to the BLOSDA unit, allowing the user to record data to a microSD card hands-free without holding the phone. The BLOSDA unit is encapsulated in skin-compatible silicone, measuring 235 × 73 × 4 mm (including straps), and is designed to be securely fastened to the wrist, as shown in Fig. 1F.

Unlike traditional stress monitoring methods that often rely on costly and bulky laboratory instruments—which are not user-friendly and require skilled personnel to operate or invasive blood tests that are uncomfortable and inconvenient, the SQC-SAS wearable offers a different solution for stress monitoring. Featuring reusable sensors for detecting MSB (cortisol) and an epidermal PSI patch for monitoring PSI signals (ECG, GSR, RAP, and ST), the SQC-SAS wearable provides a rapid, convenient, and non-invasive approach for comprehensive stress assessment. To validate the feasibility of the SQC-SAS wearable for stress monitoring, multi-modal physiological data were acquired during validated stress-inducing protocols, as well as during non-stressful conditions. Our device enables autonomous and continuous in-situ data acquisition, ensuring accurate and consistent stress evaluation (Fig. 1G). To realize stress vs. rest state detection and stressor-type classification, we also employed the Inception-MABFDNN model specifically designed to analyze stress-related patterns across various features extracted from PSI and MSB signals. This advanced machine learning (ML) model significantly improves the accuracy of stress detection and reduces the likelihood of false positives. Together, all the above position SQC-SAS wearable bioelectronic as a feasible solution for continuous and multimodal stress monitoring, offering unparalleled potential for the proactive management of stress and its associated health risks.

## Development, design, and validation of PSI patch: ECG, GSR, and RAP sensor

The ECG and GSR sensors within the PSI patch require direct skin contact to capture biopotential and bioimpedance variations[41]. Flexibility and stretchability are essential for these sensors to maintain conformal contact with the skin, accommodating any movement-induced deformations. To enhance stretchability, the PSI patch design incorporates island structures connected by serpentine-shaped patterns (Fig. 2A). When the PSI patch undergoes stretching, these serpentine ribbons can rotate within the same plane and buckle out of the plane to accommodate the applied deformation, significantly reducing strains[42]. Also, the island structure enlarges the skin contact area per unit area without enlarging the electrode size, thereby enhancing user comfort and breathability[11]. Finite element method (FEM) simulations conducted under a 20% applied strain (>skin strain tolerance ≈ 15%[43]) showed that our island-structured electrodes with serpentine interconnections (Design 1) exhibited a principal strain of only 4.7%, which is below the substrate material fracture strain limit of PI (5.1%[44]). Additionally, the maximum principal strain in Design 1 is up to three times lower than the strains observed in conventional curved ribbons, measuring 11% and 13% in Designs 2 and 3, respectively (all designs use the same island diameter of 4.32 mm and ribbon width of 250 μm) (Fig. 2C). To experimentally validate our simulation results, we measured the relative resistance change ($R/R_0$) of each electrode design under strain until the ribbons were fully ruptured. Design 3 exhibited the lowest stretchability (16%), followed by Design 2 (33%), with Design 1 demonstrating the highest stretchability (94%) (Fig. 2D). Thus, Design 1 was selected for the ECG and GSR electrodes in the PSI patch and the electrode demonstrated excellent mechanical stability in cyclic bending and stretching tests (Fig. 2E, Supplementary Fig. 2 and Supplementary Note 1). Four-island electrode design was selected to increase the contact area, showing a reduction in electrode-to-skin interface impedance and an increase in SNR up to 20.8 dB compared to designs with fewer islands (Supplementary Fig. 3).

The mechanical stability of sensors is essential for wearable devices to ensure they withstand mechanical deformations while preserving functionality. We evaluated Design 1's island-serpentine structure for stability using cyclic strain tests, subjecting it to strains from 0 to 20% at a 0.5 Hz frequency. Over 6000 cycles, the normalized resistance change remained under 2%, as shown in Fig. 2E(i), confirming the design's resilience under mechanical stress. Additionally, cyclic bending tests simulated natural skin curvature by bending the electrode from 0% to 30%, revealing similar resistance stability after 30,000 cycles (Fig. 2E(ii)). By incorporating the serpentine structure into the connections, the entire PSI patch achieved enhanced

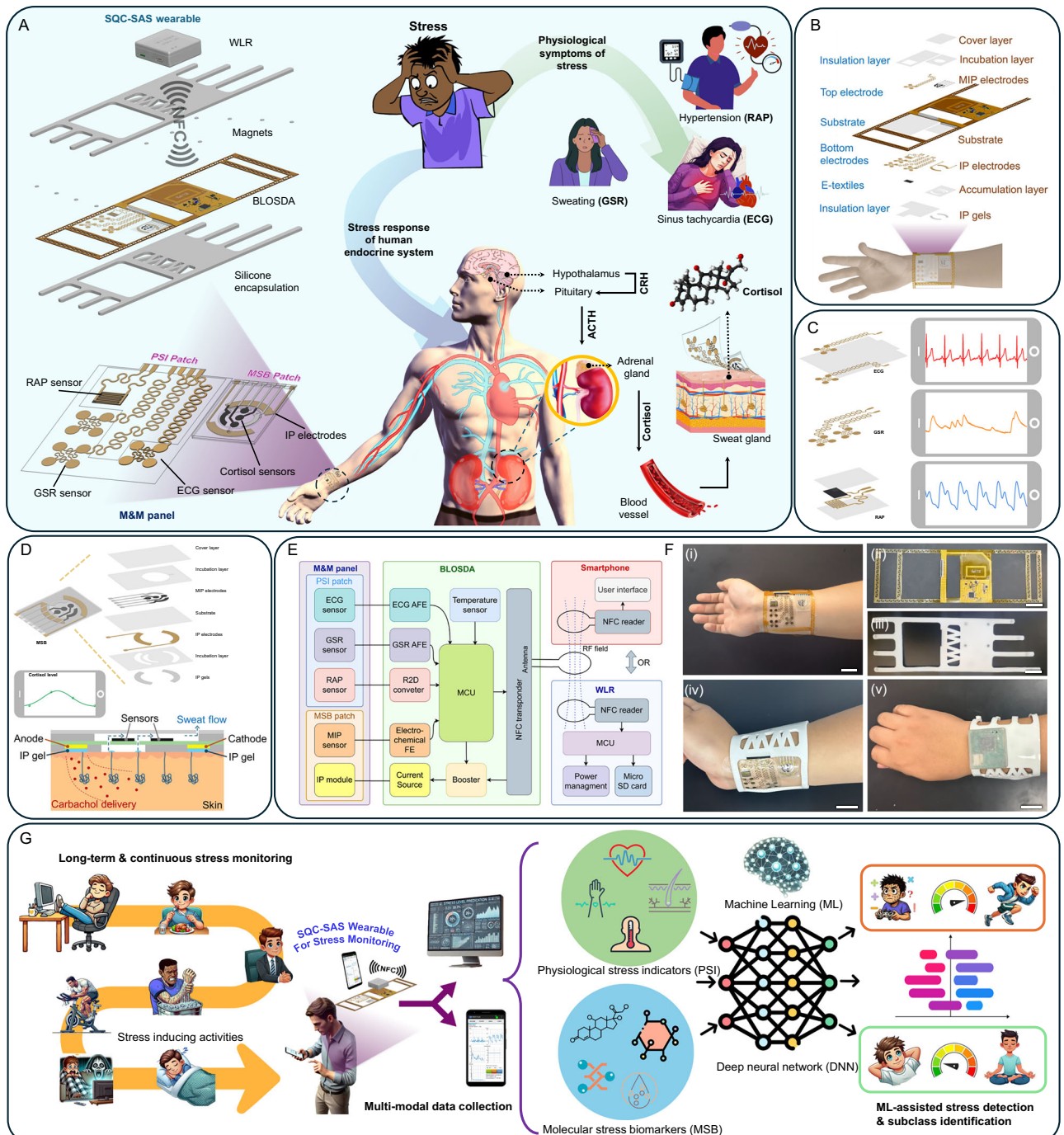

**Fig. 1 | Design and functionality of the SQC-SAS wearable. A** Conceptual overview illustrating the physiological impact of stress, the body's response mechanisms, and the role of the SQC-SAS wearable in monitoring stress-related signals. Stress stimuli trigger inflammatory responses, cardiovascular changes (e.g., heart rate, blood pressure), and sweating, mediated by cortisol release from the adrenal glands into the bloodstream, later diffusing into sweat. The wearable device, positioned on the wrist , comprises (i) the M&M panel, which integrates the PSI patch (for ECG, GSR, and RAP sensing) and the MSB patch (for cortisol detection), (ii) the flexible BLOSDA for signal acquisition from the M&M panel, and (iii) the WLR for continuous data collection and storage. **B** Layer-by-layer assembly of the M&M panel and its integration with the BLOSDA, detailing the structural organization of the system. The bottom layer contains the skin-contact electrodes for ECG, GSR, pulse sensing, and iontophoresis, while the top layer hosts the MIP electrodes for cortisol detection. **C** Design and electrode configuration of the PSI patch, highlighting the sensing structures for ECG, GSR, and RAP measurements, along with representative signal patterns from each modality. **D** Structural design and assembly of the MSB patch within the M&M panel, showing its detailed layer-by-layer composition. This includes the MIP electrode array for cortisol sensing, incubation layers for sweat collection, and a schematic of sweat extraction through wireless iontophoresis for on-demand sampling. **E** System block diagram depicting the electronic interface between the sensors, BLOSDA, and the WLR or smartphone for real-time data transmission and monitoring. **F** Optical images of the device components: (i) M&M panel showing the PSI and MSB patches on the wrist, (ii) BLOSDA without encapsulation, (iii) BLOSDA with silicone encapsulation, (iv) integrated M&M panel with encapsulated BLOSDA worn on the wrist, and (v) combined BLOSDA and WLR setup on the wrist (Scale bar: 2 cm). **G** Schematic illustration of continuous stress monitoring using the SQC-SAS wearable, capturing multimodal PSI and MSB signals to distinguish between rest and stress states, including stress subclasses. The acquired data are processed via a machine learning workflow for accurate detection and classification of stress responses.

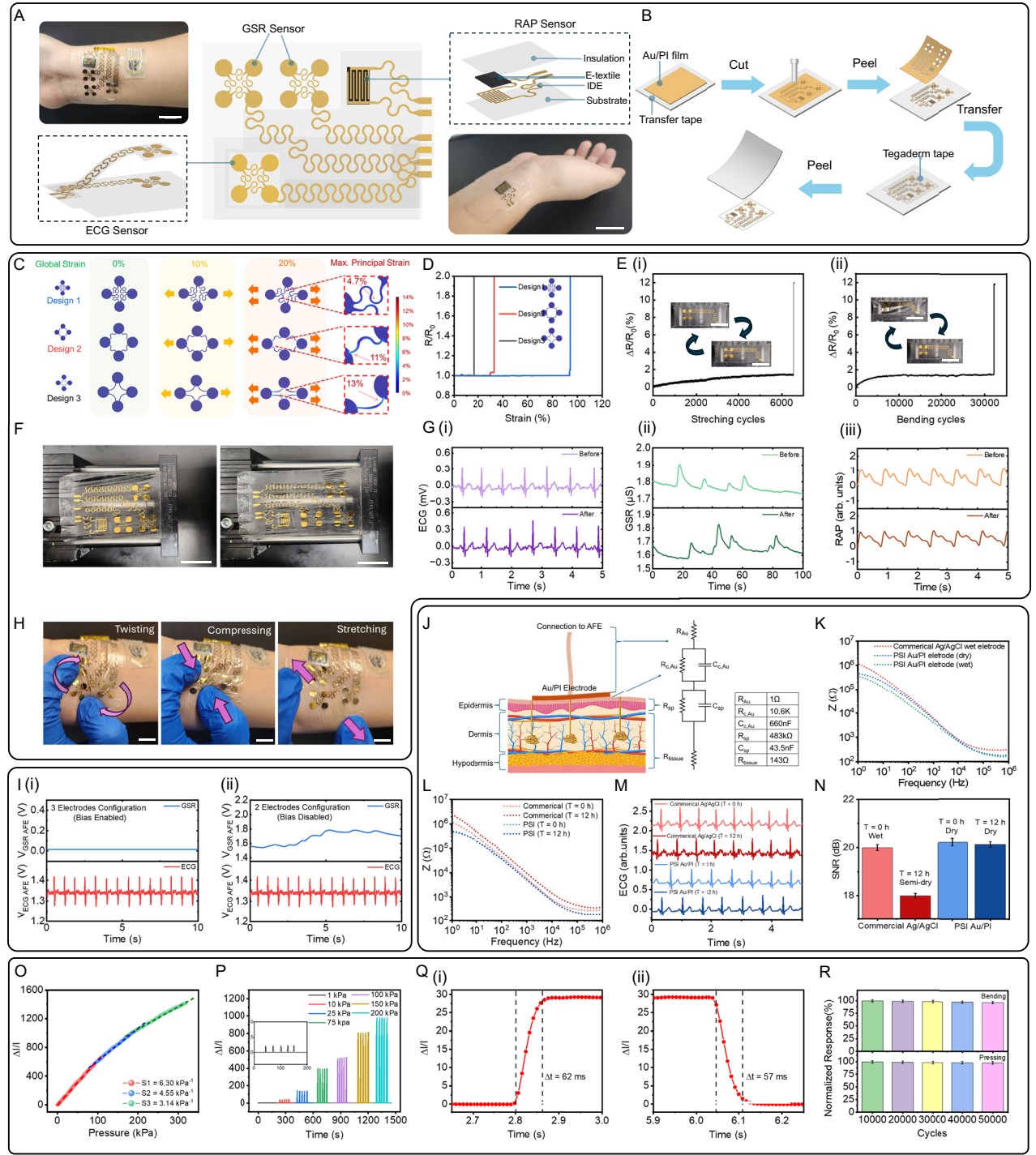

stretchability and exhibited no damage under a 30% strain, as demonstrated in Fig. 2F. As illustrated in Fig. 2G, a comparative analysis was performed on the ECG, GSR, and RAP signals collected by the PSI patch before and after it underwent 10,000 cycles of 30% bending and 2000 cycles of 20% stretching. The analysis revealed that there was no noticable degradation in signal quality subsequent to testing, thereby affirming the superior mechanical performance of the PSI panel. In addition, an on-body assessment was conducted to evaluate the PSI patch's skin conformability and mechanical integrity under various deforming conditions (Fig. 2H), such as twisting, compressing, and stretching. This assessment verified that the patch exhibited neither any noticable delamination nor any other form of damage, thus highlighting its robustness and suitability for dynamic wearable

applications. Given that both the ECG and GSR sensors are in direct contact with the skin, the conductive properties of the human body facilitate a circuit path between the two signals, necessitating careful consideration to prevent crosstalk. To address this, the bias of the ECG AFE was disabled, and a single-lead ECG configuration was employed. This setup not only minimizes crosstalk between the ECG and GSR signals but also enhances user convenience, as detailed in Fig. 2I and Supplementary Note 2.

Another critical factor in ECG and GSR electrode design is the electrode-to-skin interface impedance, which affects SNR. Traditionally, conductive gel is applied to reduce impedance and improve signal acquisition[45]; however, this method suffers from increasing impedance over time due to gel solvent evaporation during prolonged

**Fig. 2 | Design and characterization of the PSI patch of M&M panel. A** Illustration of the ECG, GSR and RAP sensors on the PSI patch and the optical image showing the PSI panel attached to skin (Scale bar: 2 cm). **B** Fabrication process for the ultra-thin Au electrodes using the CPTP method. Mechanical stability assessment of the PSI patch: **C** FEM simulation of the maximum strain distribution of the 3 electrode designs with different interconnect structures under up to 20% strain.
**D** Experimental verification of the resistance changes of the 3 electrode designs with applied strain, comparing their stretchability. **E** Relative resistance changes of the PSI panel electrodes during (**i**) 20% stretching for over 6000 cycles, and (**ii**) 30% bending for over 30,000 cycles. (Insets: photographs of the electrode during cyclic stretching/releasing and bending/releasing (Scale bar: 2 cm)). **F** Optical images of the PSI panel mounted on the test stage: (left) the initial state and (right) under 30% strain, showing no visible damage. (Scale bar: 2 cm). **G** Comparison of the (**i**) ECG (**ii**) GSR (**iii**) RAP signal collected by the PSI patch before and after 10,000 cycles of 30% bending and 2000 cycles of 20% stretching, showing no degradation in signal quality. **H** On body evaluation of the PSI patch's skin conformability and mechanical integrity during twisting, compressing, and stretching, showing no noticeable delamination or damage (Scale bar: 1 cm). **I** Crosstalk study between the ECG and GSR AFE circuits: Simulation results showing the crosstalk effects of (**i**) 3- and (**ii**) 2-electrode ECG configurations on ECG and GSR AFE outputs. (Top) Baseline shift in the GSR AFE output caused by crosstalk between ECG and GSR sensors, attributed to the bias voltage on the 3rd ECG electrode. (Bottom) GSR and ECG AFE

outputs after mitigating crosstalk by using a 2-electrode configuration for the ECG AFE, enabling simultaneous multi-sensor measurements. **J** Illustration of the cross-sectional view of the electrode-skin interface along with its equivalent circuit model, including parameter values obtained from experimental measurements. (This diagram represents only a single electrode-to-skin interface.) **K** Comparison of the skin-electrode interface impedance spectrum between our Au/PI dry and wet electrodes with commercial Ag/AgCl wet electrode demonstrating that our ultra-thin dry and wet Au/PI electrodes can achieve lower electrode-skin impedance than commercial wet Ag/AgCl electrode, suitable for long-term measurement.
**L–N** Comparison between the dry PSI Au/PI electrode and commercial wet Ag/AgCl electrode before and after 12 h of usage. (The wet Ag/AgCl electrode partially dries out after 12 h): **L** skin-electrode interface impedance spectrum and **M** ECG signals of the same subject collected through the dry PSI Au/PI electrode and commercial wet Ag/AgCl electrode before and after 12 h of usage. **N** The SNR of the ECG signals in (**M**), demonstrating the stability of dry Au/PI electrodes for long-term usage ($n = 3$ technical replicates). Data are presented as mean values ± SD. **O** Sensitivity of the RAP sensor under 0–340 kPa pressure. **P** Real-time response of the RAP sensor to applied pressure ranging from 1 to 200 kPa. **Q** (**i**) Response time and (**ii**) Recovery time of the RAP sensor. Mechanical stability tests conducted on the RAP sensor under cyclic pressing with a pressure of 10 kPa and cyclic bending at 30%:
**R** Variations in RAP sensor output responses during pressing and bending over 50,000 cycles ($n = 3$ technical replicates). Data are presented as mean values ± SD.

measurements. Our ultra-thin electrodes with serpentine connections are made from a 15 μm thick Au-coated PI film, which is thin enough (<25 μm[46]) to self-adhere to the skin surface via Van der Waals forces and form epidermal contact. We developed a cut-peel-transfer-peel (CPTP) method for low-cost rapid fabrication of conformal and stretchable ultra-thin electrodes (Fig. 2B) without the need for microfabrication. This design provides a lower skin-to-electrode contact impedance without conductive gel compared to commercial wet Ag/AgCl electrodes when applied to the skin (Fig. 2K). We hypothesize that the observed phenomenon is due to the excellent adaptability of the PSI's Au/PI electrodes, which closely conform to the microscopically uneven surface of the skin. To elucidate this interaction, we have developed a cross-sectional schematic detailing the comprehensive electrode-skin interface. As depicted in Fig. 2J, the skin is modeled as a simplified three-layer structure comprising the epidermis, dermis, and hypodermis, with eccrine sweat glands embedded within the dermis and extending to the epidermal surface via sweat ducts. The PSI Au/PI electrodes are designed to match the undulating contour of the epidermis, effectively minimizing air gaps between the Au and the skin. Based on this model, Fig. 2J presents an equivalent circuit starting with the connections from Au/PI to the various skin layers, which aids in identifying the key components influencing the electrode-skin interface impedance. Direct contact at the Au-skin interface is represented with parallel contact resistor $R_{C,\ Au}$ and contact capacitor $C_{C,\ Au}$. The stratum corneum within the epidermis is depicted with a parallel resistance and capacitance model ($R_{sp}$ and $C_{sp}$), while the dermis and hypodermis are modeled as a resistor ($R_{tissue}$). Estimated values of these parameters are derived from measured skin-electrode contact impedance spectra and are listed in Fig. 2J. To further highlight the advantages of the dry PSI Au/PI electrode, we demonstrate its ability to maintain lower contact impedance compared to commercial Ag/AgCl gel electrodes, making it more suitable for long-term measurements. Figure 2L compares the contact impedance of both the dry PSI Au/PI electrode and the commercial Ag/AgCl gel electrode before and after 12 h of usage. The dry PSI Au/PI electrode exhibited negligible changes in skin-electrode contact impedance, whereas the commercial Ag/AgCl gel electrode displayed increased impedance over time due to gel dehydration. Additionally, we assessed the ECG signals collected with these electrodes (Fig. 2M). The results indicate that the increased impedance in the commercial electrode led to a decreased SNR, while the PSI Au/PI electrode showed no noticeable degradation in ECG signal quality (Fig. 2N).

The RAP sensor within the PSI patch needs to detect sub-kilopascal pressure variations induced by the RAP, which is achieved by our resistive pressure sensor. The sensor comprises an e-textile laid over IDE, which is sandwiched between insulation and substrate layers (Fig. 2A). E-textile was made by coating conductive material on a woven fabric. This design relies on the working principle that the inter-contact area within the woven mesh structure of the e-textile expands under pressure, establishing more conductive pathways and thereby enabling RAP monitoring through conductance changes[47]. Cotton fabric was chosen for e-textiles not only for its woven mesh structure but also for its flexibility, lightweight, cost-effectiveness, and ease of coating. The MXene/PANI/PEDOT composite was selected specifically for its ability to enhance the textile's conductive properties while maintaining flexibility and durability essential for wearable applications. This composite leverages the complementary properties of MXene, PANI, and PEDOT, achieving superior conductivity and mechanical resilience. MXene/PANI/PEDOT was synthesized as the conductive material due to the interaction between MXene, PANI, and PEDOT, driven by π-π bonding and electrostatic forces, enhancing conductivity[48]. Raman spectroscopy was also performed and confirmed the successful synthesis of MXene, PANI, PEDOT, and the MXene/PANI/PEDOT composite (Supplementary Fig. 5). Additionally, SEM images of the e-textiles show that the MXene nanosheets, PANI, and PEDOT uniformly adhered to the cotton fabric surface via the dip-coating method (Supplementary Figs. 4 and 5). The e-textile coated with a 1:1:1 ratio of these materials resulted in optimal conductance of $1.4 \times 10^5$ S compared to other combinations (Supplementary Fig. 5 and Supplementary Note 3). The IDEs under the e-textile carry the excitation signal from the BLOSDA to the e-textile, enabling the conversion of applied pressure into an electrical signal. The sensitivity of our resistive-based RAP sensor was evaluated by applying a constant excitation voltage and monitoring the relative current response ($\Delta I/I_0, \Delta I = I - I_0$) while applying external pressure ($P$) ranging from 0 to 340 kPa. The sensitivity of the RAP sensor ($S$) is calculated as $S = (\Delta I/I_0)/\Delta P$. The sensitivity of the sensor varies across pressure ranges, showing values of 6.30, 4.55, and 3.14 kPa$^{-1}$ for the ranges 0–100, 100–200, and 200–340 kPa, respectively (Fig. 2O). The reduction in sensitivity at pressures above 100 kPa is attributed to the densification of voids within the e-textile[49]. Nonetheless, the results indicate that the sensor's sensitivity across all evaluated pressure ranges remains sufficiently high for effective RAP sensing. The sensor's repeatability was tested by applying constant pressures from 1 kPa to

200 kPa over five cycles, with results indicating a stable and repeatable relative current response at each pressure level (Fig. 2P). The MXene/PANI/PEDOT-based RAP sensor shows a rapid response time of 62 ms and a recovery time of 57 ms, sufficient for real-time RAP monitoring (Fig. 2Q). The RAP sensor demonstrated high mechanical stability and flexibility, maintaining consistent performance with <5% and <7% variation over 50,000 pressing and bending cycles, respectively, confirming its reliability for continuous wearable use on skin subject to various deformations (Fig. 2R, Supplementary Fig. 6, and Supplementary Note 3).

## Assessment, characterization, and optimization of the MSB patch

The design and fabrication of the MSB sensor patch within the SQC-SAS wearable involves careful selection of the substrate and electrode materials to ensure optimal performance for on-skin applications. The substrate for the MSB sensor patch was fabricated using SEBS, selected for its excellent biocompatibility with human skin and mechanical properties closely resembling those of Tegaderm tape, the substrate used for the PSI patch (Supplementary Fig. 7A). SEBS exhibits a Young's modulus closely matching that of human skin (0.1–2.4 MPa)[50] and offers exceptional flexibility and stretchability of up to approximately 500%. This significantly outperforms commonly used materials in flexible devices, such as polyimide (PI) and polyethylene terephthalate (PET), which have substantially higher Young's moduli and lower fracture strains (PI: 2.23 GPa, 50%; PET: 2.79 GPa, 70%, respectively)[51], making SEBS a more suitable choice for conformal and stretchable wearable applications. While the MSB substrate gains stretchability from SEBS, the MSB electrodes must also be stretchable, durable under bending and stretching, and remain conductive. To meet these requirements, we combined Ag with SEBS, producing flexible and conductive electrodes with an electrical conductivity of $10^4$ S/cm (Supplementary Fig. 8). Next, MIP layers were integrated into electrode arrays of the MSB sensor patch for their ability to replicate the binding properties typically associated with antibodies or enzymes, enabling precise detection of targeted sweat cortisol molecule levels with exceptional sensitivity and selectivity[52]. A major challenge in developing MIP-based sensors is achieving environmental stability, restoration, and reusability for continuous, real-time monitoring in a wearable format. Some recent approaches have involved off-site sensor restoration using specific washing solutions[53]. However, this method risks compromising the MIP structure, potentially causing polymer swelling, shrinking, or loss of functional groups, which can reduce the sensor's selectivity and affinity for target molecules[54]. Additionally, using external wash buffers is impractical for wearable applications. In our research, we addressed these challenges by designing MIPs for restoration without the need for external washing solutions. This was achieved by applying an overpotential voltage through the electrochemical chronoamperometry (ECA) technique. MIPs typically face overoxidation issues at higher potentials, resulting in the formation of functional groups[55] that could adversely affect their structure and limit their reusability[56]. To overcome this problem, we used pyrrole as a functional monomer for MIP synthesis. During polymerization, these reactive species of monomer underwent electrophilic substitution reactions, forming covalent bonds between the targets. This approach ensured that after polymerization, the functional groups of MIPs became embedded within the polymeric structure on the electrode arrays of the MSB. Subsequent extraction of the target molecules revealed binding sites or imprinted cavities on the MIP-modified electrode that were designed to be complementary in terms of size, shape, and charge to the target analyte, thus achieving highly selective detection of the target analyte (Fig. 3A). In contrast, non-imprinted polymer (NIP) layers lack these binding cavities and are not affected by the presence of target biomolecules. We examined the surface morphology through SEM to verify the formation of imprinted cavities in

the MIPs. The SEM images reveal the presence of pores within the MIP layer and the absence of pores on the NIP surface (Fig. 3B, C inset). Further, the quantification of cortisol molecules in sweat relies on measuring the oxidation of target molecules bound to the MIP template using ECA. The current observed in the ECA measurement directly correlates with the concentration of the target analyte. We investigated the detection of biomolecules in a phosphate-buffered solution (PBS) solution at various concentrations using the MSB's electrode array. To achieve this, we prepared different concentrations of cortisol molecules individually in a 0.1 M PBS solution, ranging from 1 nM to 10 μM, encompassing the physiological range found in sweat[36]. Subsequently, we utilized the fabricated MIP and NIP-modified MSB electrode arrays to examine the various concentrations of the cortisol molecules (Fig. 3B, C). When employing the MIP-modified MSB's electrode arrays for cortisol detection, we observed a linear decrease in current response concerning the logarithmic concentration of cortisol (Fig. 3D), achieving a sensitivity of 0.2447 μA/log[nM] and a high correlation coefficient ($R^2 \approx 0.993$). This decrease in current on the MIP-modified sensor indicates the specific binding of cortisol molecules within the imprinted cavities. This specific binding inhibits the electron transfer pathways of PB, leading to a reduction in PB oxidation current. This variation in current was further corroborated using cyclic voltammetry (CV) (Supplementary Fig. 9). The integration of the PB redox transducer into the imprinted polymer and the tuning of experimental settings allow for rapid, wearable ECA biomolecule sensing at a low voltage of +0.1 V, as determined by CV. In contrast, the control experiment utilizing NIP (Fig. 3D) exhibited a negligible response across the same cortisol concentration range, with a sensitivity of only -0.005 μA/log[nM]. These results indicate the absence of biomolecule-binding cavities within the NIP layer. As depicted in Supplementary Fig. 10 and Supplementary Note 4, to determine the optimal incubation time that allows cortisol binding to the MIP layer, we tested incubation periods from 5 s to 5 min. Using a 100 nM cortisol solution and ECA measurements at +0.1 V, we observed a rapid decrease (-50%) in current as incubation time increased, indicating increased cortisol binding. This response stabilized after 1 min, establishing it as the minimum incubation time for all subsequent measurements (Supplementary Fig. 10A). Selectivity is another vital factor in obtaining accurate results from sweat samples. The MIP-modified MSB's electrode arrays not only provide selective detection of target biomolecules but also efficiently discriminate against a wide range of common sweat components that may interfere during the transduction process. Figure 3E demonstrates the sensor's current response after incubation with various common interfering species of progesterone and cortisone at 1 μM concentration in PBS and subsequent introduction of the corresponding 1 μM cortisol molecules in PBS. The MIP sensor exhibited a distinctive response to cortisol, suggesting highly selective detection even in the presence of an excess of potential interfering species, where no response was observed when only interfering species were present. To further assess our sensor's applicability, we evaluated the performance of the MIP sensor in an AS environment. An artificial sweat (AS) solution with varying cortisol concentrations (1 nM to 10 μM) was prepared to simulate real conditions and test the sensor's capability to detect cortisol in sweat samples (Supplementary Fig. 10B). The resulting current response displayed a linear relationship with the logarithm of cortisol concentration (Supplementary Fig. 10C), with a sensitivity of 0.24 μA/log [nM] ($R^2 \approx 0.99$), closely matching the PBS results.

The primary challenge in modern sensing technologies for wearable, long-term monitoring is sensor reusability (Fig. 3F). To address this, we developed a restoration process designed to restore the initial current value after removing all immobilized biomolecules adhering to the MIP matrix(Fig. 3K). We began by optimizing the restoration process through an evaluation of various electrochemical voltammetry methods, including differential pulse voltammetry (DPV),

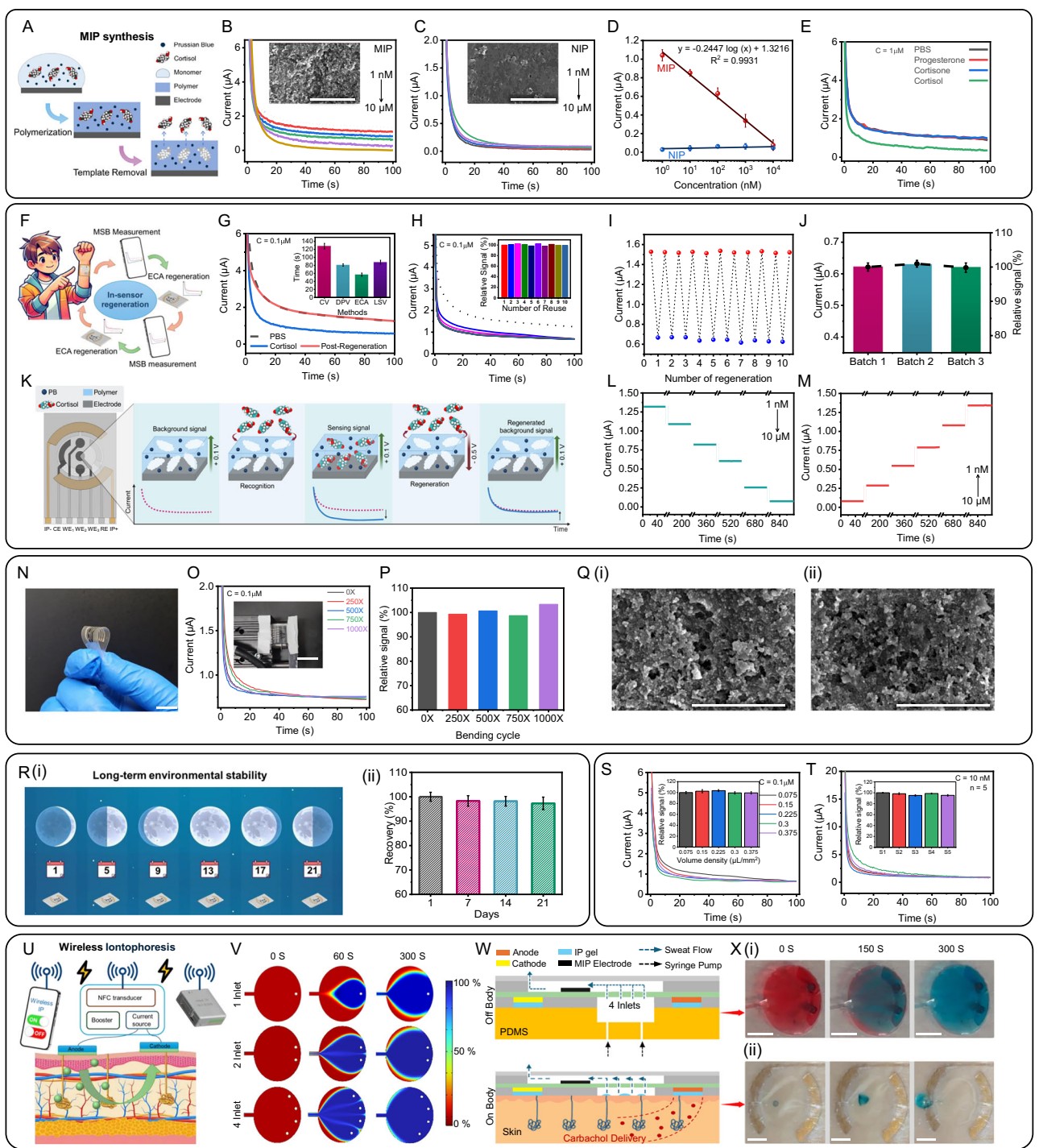

linear sweep voltammetry (LSV), ECA, and CV, to identify the method capable of achieving a complete background current recovery ratio in the shortest possible time. As shown in Fig. 3G inset, ECA demonstrated the best performance in terms of restoration time, achieving a full recovery ratio in approximately 55 s, whereas other methods required 80–130 s. Consequently, ECA was chosen for subsequent restoration experiments. Figure 3G shows the current response of the MIP sensor to different conditions: background (PBS without cortisol), sensing signal (PBS with 100 nM cortisol), and post-restoration (PBS without cortisol again). The complete recovery of the background current after restoration confirms the sensor's capability for multiple reuses, validating the effectiveness of the restoration process. To confirm the repeatability of our restoration process, we evaluated the reusability of our MIP sensor across multiple restoration cycles. As

illustrated in Fig. 3H, I, the ECA results from the reused MSB electrode arrays demonstrate reliable sensor performance after ten restoration cycles. The sensors exhibited consistent current signals when exposed to the same sample, with a relative standard deviation (RSD) of less than 5%, underscoring the robustness of the restoration process and sensor reusability. Supplementary Movie 9 demonstrates the ex-situ restoration process of the reusable MSB patch for continuous cortisol measurement in AS samples, using the same sensor and delivering accurate, repeatable results across several consecutive measurements. To assess the repeatability and reliability of the MIP sensor fabrication process, we conducted a batch-to-batch consistency analysis. Specifically, we fabricated three batches of MSB electrode arrays and evaluated the current response of each batch to cortisol molecules in 0.1 M PBS at a concentration of 100 nM. As shown in Fig. 3J, all

**Fig. 3 | Design, optimization, and characterization of MSB patch of M&M panel.**
**A** Schematic of the PPy-MIP preparation procedure on SQC-SAS wearable's MSB electrodes. ECA response of the MIP (**B**) and NIP (**C**) cortisol sensor to different cortisol concentrations (1 nM–10 μM) in PBS. (Insets: SEM images of MIP and NIP, respectively (Scale bar: 5 μm); $n = 3$ technical replicates). **D** Corresponding calibration plots of the MIP (red) and NIP (blue) cortisol sensors in PBS. MIP sensor shows a logarithmic response of the electrode current to the cortisol level. **E** Selectivity of the MIP cortisol sensor, demonstrated by the ECA responses of the sensor to 1 μM cortisone, progesterone, and cortisol in PBS, highlighting its specificity for cortisol detection ($n = 3$ technical replicates). Data are presented as mean values ± SD. **F** Illustration of the device's restoration capability, enabling the reuse of the sensor. **G** ECA response of the MIP cortisol sensor before and after the restoration process measured in PBS. (Inset: Optimization of the restoration process showing the time taken by the different voltammetry techniques (CV, DPV, ECA, and LSV) to reach 100 % recovery; ($n = 3$ technical replicates). Data are presented as mean values ± SD. **H** ECA response of the MIP cortisol sensor to same cortisol concentration in PBS after multiple restoration showing the reusability of the sensor. **I** Demonstration of the restoration and repeated sensing of the same cortisol concentration in PBS using the same MSB sensor multiple times, highlighting the sensor's reusability and stability. **J** Batch-to-batch consistency of restored MSB sensors from three separate batches, each containing three sensors ($n = 3$ technical replicates), demonstrating reliable and reproducible sensing of the same cortisol concentration. **K** Schematic illustration of cortisol biomolecule detection and restoration processes using MIP. ECA measurement of MSB sensor

from lower to higher concentration (**L**) and higher to lower concentration (**M**) of cortisol in PBS. **N** Optical images of the MSB patch demonstrating its flexibility (Scale bar: 1 cm). **O** ECA response of the MSB sensor during 1000 cycles of 30% bending (Inset: Optical image showing the flexibility of the MSB sensor under 30% bending (Scale bar: 2 cm)). **P** Bar plots showing the corresponding relative signal retention over 1000 cycles. **Q** SEM image of MIP working electrode (Scale bar: 3 μm; $n = 3$ technical replicates) (**i**) before and (**ii**) after the 1000 bending cycles test. **R** (**i**) Illustration of the long-term environmental stability of the MSB sensor. (**ii**) Experimental results demonstrating the environmental stability of the MSB sensor, showing more than 96% retention of current response over a 3 weeks testing period ($n = 3$ technical replicates). Data are presented as mean values ± SD. **S** Effect of sample volume on the ECA response and relative output current (inset; $n = 3$ technical replicates) of the MSB cortisol sensor. **T** Overlaid ECA response and output current (inset) of five independent MSB sensors ($n = 5$) responding to 10 nM cortisol, demonstrating sensor-to-sensor reproducibility. **U** Illustration of the sweat sample collection process using the NFC-powered wireless IP system of the SQC-SAS wearable. **V** Simulated effect of microfluidic inlet number on channel refreshing rate at different time stamps. **W** Illustration of the microfluidic channel during off-body flow test for sample refreshing and the on-body validation of IP sweat stimulation and sweat sampling. **X** (**i**) Optical image of a microfluidic channel where the old sample with red dye is refreshed with the new sample with blue dye (Scale bar: 5 mm). (**ii**) Optical image of the wireless IP sweat-inducing and collection using the MSB patch. MIP electrodes were removed, and dyes were added to the sample reservoir to aid in the visualization of sweat flow (Scale bar: 5 mm).

batches exhibited consistent signals with an RSD of less than 3%, demonstrating the repeatability of our fabrication process. We further validated the repeatability of our MIP sensor by measuring its current response across a range of cortisol concentrations in PBS, from blank (0 nM) to 10 μM, in both ascending and descending orders. As shown in Fig. 3L, M, the MIP sensor consistently provided accurate detection of cortisol across this broad concentration range, demonstrating its stability and precision across different concentration sequences.

Ensuring mechanical stability is also essential for the reliability of skin-worn sensors, particularly under bending deformations encountered during wear (Fig. 3N). To evaluate the impact of bending on sensing performance, the MSB sensor was subjected to 30% cyclic bending for up to 1000 cycles (Fig. 3O). After every 250 cycles, the ECA signal of the MSB sensor in the same buffer solution was measured and the results revealed a high retention rate of approximately 98.3%, demonstrating the device's resilience and consistent functionality under mechanical stress (Fig. 3P). The SEM images shown in Fig. 3Q and Supplementary Fig. 11 show no visible signs of peeling, cracking, or delamination of the MIP and Ag reference electrodes before and after the 1000 cyclic bending cycles. Long-term stability is another crucial factor in wearable sensors' practical applications. The MSB electrode array was evaluated for long-term stability over a 3 weeks period under ambient conditions (Fig. 3R and Supplementary Note 4). Notably, variations in sample volume (0.075–0.375 μl/mm²) did not produce substantial changes, as depicted in Fig. 3S. This stability makes it possible to achieve accurate cortisol detection even with low sweat volumes. Furthermore, MSB sensors within the same batch showed highly reproducible results, with an RSD of less than 2%, further confirming the robustness of the fabrication process (Fig. 3T and Supplementary Note. 4).

The integration of the MSB electrode array with a microfluidic channel and a wireless IP system enables on-demand sweat management and in-situ cortisol detection. The use of carbachol iontophoresis gel was first confirmed to be both biocompatible and effective for sustained sweat secretion, making it highly suitable for continuous wearable applications (Fig. 3U, Supplementary Fig. 1 and Supplementary Note 4). The microfluidic design allows for efficient low-volume sampling and precise separation of sweat sampling areas from IP gels, minimizing contamination and preventing sample evaporation. As illustrated in Fig. 3U the wireless IP system can be remotely controlled

and powered by a smartphone and WLR, providing a fully wearable sweat stimulation solution, unlike existing market alternatives that require an external power source. For this, firstly, we employed numerical simulations to optimize the microfluidic channel's design, including the arrangement of inlets, as this impacted sample collection and refreshing efficiency (Fig. 3V, Supplementary Fig. 12 and Supplementary Note 5). The simulation results underscored the enhanced refreshment rate associated with an increased number of inlets. Thus, we used a 4-inlet microfluidic channel for subsequent experiments. The functionality of the microfluidic channel was further validated through both in vitro and in vivo flow tests (Fig. 3W, X). As demonstrated in Supplementary Movie 1, the empty top incubation channel was completely filled with fresh samples from the bottom accumulation layer during the in vitro flow test without any bubble formation, confirming its efficient fluid dynamics. Additionally, another in vitro flow test (Fig. 3X(i) and Supplementary Movie 2) showed that a channel previously filled with an old sample was refreshed with a new sample in 300 s, ensuring complete sample replacement. The in vivo flow test was performed alongside the verification of the IP system, which plays a crucial role in enabling sweat stimulation wirelessly. By eliminating the need for external power sources or batteries, the wireless IP system enhances the device's portability and usability. To validate its functionality within the MSB sensor patch, an experiment was conducted to assess its capability to stimulate and sustain sweat production for real-time biochemical monitoring. For this, an MSB panel without MIP electrodes was used, and blue dye was placed in the sensing chamber. The MSB sensor patch was then connected to the BLOSDA and attached to a subject's hand. By positioning a mobile phone near the NFC antenna of the BLOSDA system, the wireless IP was activated, delivering the IP current for 5 min. The chamber's 8.4 μL capacity filled with sweat in the next 5 min, flushing out the blue dye, as shown in Fig. 3X(ii) and Supplementary Movies 3 and 4. This experiment demonstrated the wireless IP system's effectiveness for localized, on-demand sweat induction and sampling, underscoring the MSB sensor patch's capability for continuous molecular sensing.

**In vivo verification of the M&M panel.** To confirm that the sensors on the M&M panel can reliably provide biosignals for subsequent stress analysis, we assessed the device's performance on human subjects. Figure 4A displays the on-body ECG signals captured using the PSI

patch, highlighting the extraction of key morphological features, the PQRST complex[57], depicted in Fig. 4B. Similarly, Fig. 4C, D presents the on-body GSR signals obtained with the PSI patch, illustrating the extraction of essential morphological features, specifically the Tonic and Phasic components[58]. Figure 4I, J details the RAP signals measured from the human wrist using the RAP sensor, featuring an enlarged view of a single pulse cycle that distinctly captures critical morphological characteristics, including P-, T-, and D- waves[59]. These results demonstrate that the sensors on the M&M panel exhibit high sensitivity and precision, providing informative biosignals that enable the successful extraction of critical features necessary for training ML models to accurately classify stress events. The dynamic monitoring of cortisol level variations throughout daily activities and the rapid assessment of cortisol responses to stimuli are crucial for evaluating both chronic and acute stress levels in individuals. To demonstrate the practical application of the MSB patch for long-term, on-site, near real-time, and semi-continuous analysis of sweat cortisol levels, we conducted an in situ proof-of-principle study focusing on cortisol circadian rhythms.

The MSB patch, integrated with the IP and microfluidic systems, was used to track cortisol level fluctuations in human sweat at 10 a.m. and 5 p.m., chosen to capture the typical peak in the morning and decline in cortisol levels throughout the day. Sweat induction, cortisol measurement, and sensor restoration followed the optimized procedures outlined above. Figure 4K shows the cortisol concentrations for different subjects, recorded in the morning and afternoon on the same day. The cortisol concentration showed a decrease from morning to night, consistent with the natural circadian reduction in cortisol levels[60]. Supplementary Movie 10 demonstrates wireless IP, simultaneous PSI, and MSB signal collection, and the restoration process of the reusable MSB patch for continuous stress monitoring on the human wrist using the SQC-SAS wearable, delivering stable results across two consecutive measurements. To demonstrate the sensor's reusability and stress monitoring capability, we utilized the MSB patch to measure sweat cortisol levels before and after administering the Stroop test, which serves as a stress inducer. Figure 4L displays the sweat cortisol concentrations for different subjects, recorded before and after the Stroop test. The data indicate an increase in cortisol levels following the stress event, aligning with the typical endocrine system response to stress[61]. These findings highlight the MSB patch's effectiveness in accurate, on-demand monitoring of sweat cortisol and stress detection.

The SQC-SAS wearable's M&M panel was further evaluated by comparing it with commercial gold-standard systems. PSI and MSB signals were captured simultaneously from the same subject using both the M&M panel and the commercial devices, facilitating a direct comparison of performance. To ensure the accuracy and reliability of the physiological signals captured by the M&M panel, two evaluation processes were devised, using the commercial systems as a reference. First, the R-squared ($R^2$) score was established as the metric for evaluation. This metric acts as a dependable indicator to identify any morphological variations between the signals captured by the panel and those captured by commercial systems. Figure 4E, G, M illustrates example samples of ECG, GSR, RAP, and cortisol signals recorded by the commercial systems and the M&M panel simultaneously. The initial visualization demonstrates that the M&M panel can capture physiological signals with quality comparable to commercial systems. Computed $R^2$ score values of 0.98, 0.99, and 0.98 for ECG, GSR, and RAP indicate that the M&M panel successfully captures the morphological characteristics of all physiological signals. The MSB patch was further compared with commercial gold-standard systems, like enzyme-linked immunosorbent assay (ELISA) for cortisol detection. Figure 4O displays the cortisol concentration of four human sweat samples measured by two systems. The regression analysis reveals an $R^2$ value of 0.99, indicating a high level of correlation between the results from our device and the gold-standard measurement method.

Second, the integrity of each signal was evaluated by extracting the most crucial feature from each signal captured with the M&M panel and comparing them with the same feature from signals recorded simultaneously by commercial systems. This comparison was conducted using the Bland-Altman plot[62–67], which assesses the agreement between two different measurement techniques by plotting the difference between the measurements against their mean, evaluating the relationship between the extracted features from signals recorded by the M&M panel and those recorded by commercial systems. Figure 4F, H, N, P showcases the Bland-Altman plots for all physiological signals. According to these plots, the mean error difference between the features extracted from the M&M panel and the commercial signals are 0.71429 for heart rate (HR) derived from ECG, 0.88571 for GSR phasic peaks count, −0.03044 for RAP systolic peaks amplitude, and 0.00146 for the cortisol concentration. These values indicate that the average difference between the features extracted from the M&M panel and commercial signals is quite minimal. The results demonstrate that the sensors on the SQC-SAS exhibit reliable performance comparable to commercial products.

To further validate the biocompatibility of the M&M panel, we conducted a series of evaluations focusing on the Au/PI electrode and IP gel, materials that are in direct contact with the skin. Other components of the panel utilized commercially available medical tape. The viability of normal human dermal fibroblast cells grown on these materials was assessed using calcein-AM/EthD-1 staining at days 1, 3, and 7 of culture (Fig. 4Q, R, S, U). Additionally, the metabolic activity of the normal human dermal fibroblast cells was analyzed to provide a comprehensive view of cellular health and compatibility (Fig. 4T, V). We further compared the biocompatibility of our IP gel (Carbagel), which incorporates carbachol, with the commercially available Pilogel. Details of this comparison are discussed in Supplementary Note 4 and illustrated in Supplementary Fig. 1. These assessments ensure that all materials directly interacting with the skin are safe and non-toxic, further substantiating the panel's suitability for prolonged contact during wearable applications. To demonstrate the long-term stability, reusability and reliability of the SQC-SAS system, we conducted continuous physiological monitoring across 14 days. The physiological signals ECG, GSR, and RAP were consistently recorded from day 1 through day 14, showing minimal degradation in signal quality (Fig. 4W, X, Y). Daily sweat cortisol levels were tracked from 8:00 to 20:00 h, maintaining stable readouts over the same period and the results further validate the daily cortisol circadian rhythm (Fig. 4Z). These results underscore the system's stability, reusability and its ability to support prolonged sensing.

## Comprehensive evaluation and performance of the SQC-SAS wearable for continuous stress monitoring

The SQC-SAS wearable integrates both PSI and MSB sensing functionalities, enabling continuous multimodal signal recording (Fig. 5A). The MSB sensor is designed for cortisol monitoring, with its layered construction effectively separating sweat sampling areas from the IP gels to ensure accurate sweat collection and analysis. This design enables precise control of sweat flow and maintains sample integrity, as illustrated in Supplementary Fig. 13. Concurrently, the PSI patch incorporates essential sensors for monitoring PSIs. The PSI patch's ultra-thin electrodes, laminated onto Tegaderm film and fabricated using the CPTP method (Supplementary Figs. 14, 15 and Supplementary Note 6), exhibit excellent mechanical durability and flexibility. This design enables the electrodes to conform seamlessly to non-flat surfaces, such as the human wrist, without damage, ensuring reliable performance even under dynamic conditions. To facilitate on-body continuous and long-term monitoring, the M&M panel is integrated with the SQC-SAS wearable's BLOSDA. This integration enables wireless data collection and IP for on-demand sweat sampling, powered and controlled by the WLR or smartphone through NFC (Fig. 5B and

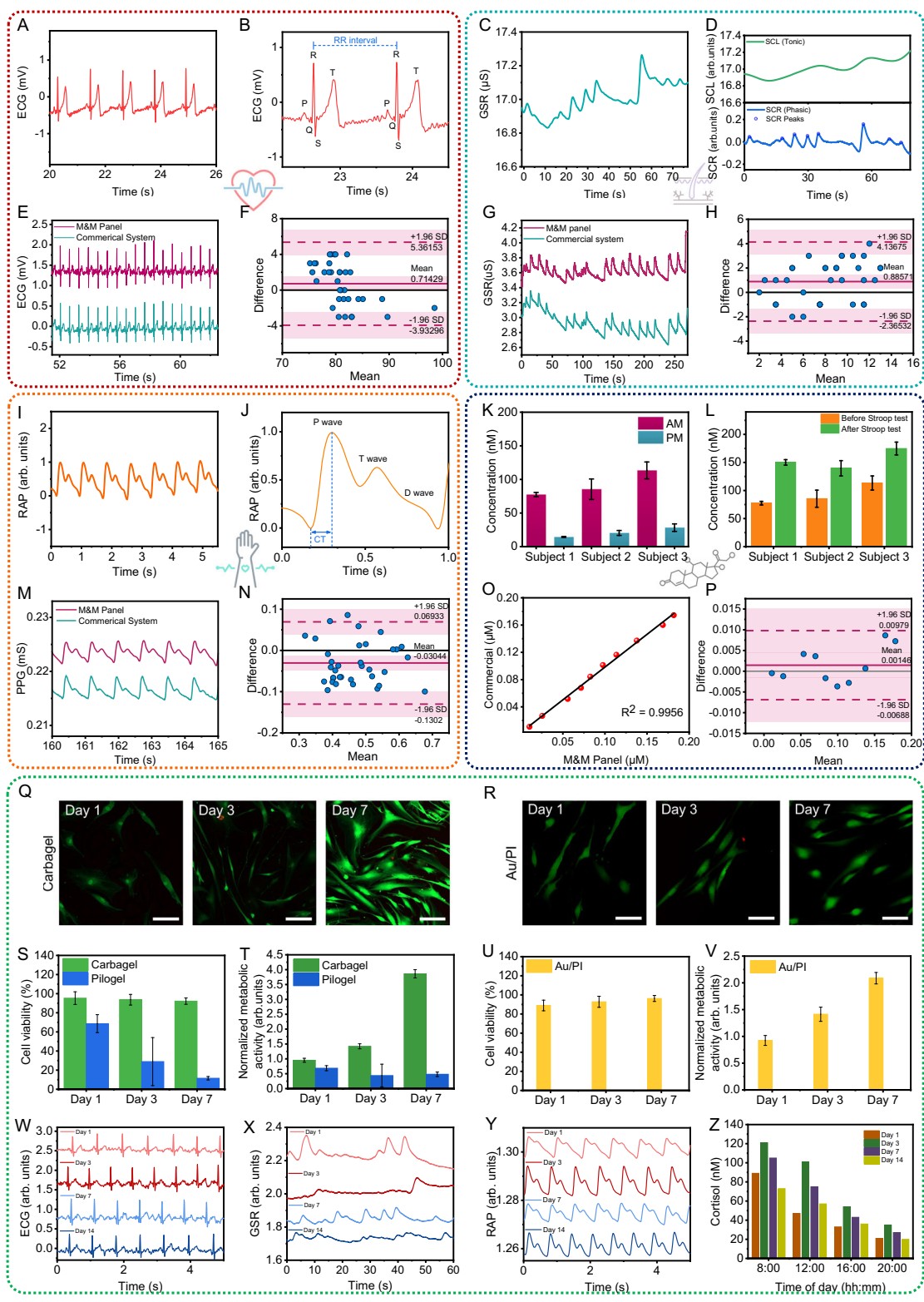

Supplementary Note 7). The flexible BLOSDA integrates an NFC transponder (tag) that harvests electromagnetic energy from the NFC reader through a flexible antenna. The system includes MCU and distinct AFE circuits, which transduce raw signals from the M&M panel into digitized form for wireless data transmission. Additionally, the BLOSDA features an integrated IP current source, enabling NFC-powered IP for on-demand sweat stimulation. The wireless power harvesting capability eliminates the need for a battery, thus offering BLOSDA enhanced wearability and flexibility. This design achieves

virtually infinite battery lifetime, enabling indefinite operation as long as WLR or smartphone remains nearby. The app developed for smartphones can configure SQC-SAS wearable and virtualize multi-sensor data in real-time. (Fig. 5C, Supplementary Movies 5 and 7). The WLR's primary functions include wireless power transmission to BLOSDA and continuous data reading, facilitated by an MCU, NFC reader, and flexible antenna, all enclosed in a 3D-printed case. Collected data is stored on a microSD card for further processing, enabling continuous data collection during daily activities and

**Fig. 4 | In vivo verification of the M&M panel. A** Real-time ECG signal obtained from the PSI patch. **B** The P, Q, R, S, T peaks of the ECG waveform and RR-interval, showing the ability to accurately capture corresponding signal patterns. **C** Real-time GSR signal obtained from the PSI patch. **D** The Tonic (SCL) and phasic (SCR) components of GSR signal extracted from (**C**). **E** The comparison between the ECG signals collected by the SQC-SAS wearable and commercial ECG sensor. **F** The Bland–Altman plot showing the correlation between the HR analyzed from the ECG signals collected by the SQC-SAS wearable and commercial ECG sensors. **G** The comparison between the GSR signals collected by the SQC-SAS wearable and commercial GSR sensor. **H** The Bland–Altman plot showing the correlation between the SCR peaks analyzed from the GSR signals collected by the SQC-SAS wearable and commercial GSR sensors. **I** Real-time RAP signal obtained from the wrist using the PSI patch. **J** The P, T, D wave and CT of the RAP waveform, demonstrating the ability to accurately capture corresponding signal patterns. **K** The corresponding sweat cortisol concentrations of different subjects measured by the MSB patch in the morning and evening, following the circadian rhythm of cortisol ($n$ = 3 technical replicates). Data are presented as mean values ± SD. **L** The corresponding sweat cortisol concentrations of different subjects measured by the MSB patch before and after Stroop test, shown an increase in cortisol after stress induction ($n$ = 3 technical replicates). Data are presented as mean values ± SD. **M** The comparison between the RAP signals collected by the SQC-SAS wearable and

commercial PPG sensor. **N** The Bland–Altman plot showing the correlation between the peak amplitude analyzed from the RAP signals collected by the SQC-SAS wearable and commercial PPG sensors. **O** The comparison between the on-body cortisol concentrations collected by the SQC-SAS wearable and commercial Cortisol ELISA kit. **P** The Bland–Altman plot showing the correlation between the cortisol concentrations analyzed from the SQC-SAS wearable and commercial Cortisol ELISA kit. Image representatives of live/dead fluorescent micrographs of normal human dermal fibroblast cells cultured at 1, 3, and 7 days in direct contact with **Q** Carbagel of the MSB patch and **R** Au/PI electrode of the PSI patch (Scale bar: 50 μm). Biocompatibility comparison between Carbagel and Pilogel: **S** viability quantification of normal human dermal fibroblast cells assessed by calcein-AM/ EthD-1 on days 1, 3, and 7 of culture and **T** Metabolic activity assay on days 1, 3, and 7 of culture of normal human dermal fibroblast cells ($n$ = 3 technical replicates). Data are presented as mean values ± SD. Biocompatibility of the Au/PI electrode: **U** viability quantification of fibroblast cells assessed by calcein-AM/EthD-1 on days 1, 3, and 7 of culture and **V** Metabolic activity assay on days 1, 3, and 7 of culture of fibroblast cells ($n$ = 3 technical replicates). Data are presented as mean values ± SD. Long-term stability of the SQC-SAS for 14 days demonstrating reusability of the sensors and the capability of prolonged sensing **W** ECG, **X** GSR, **Y** RAP signals and **Z** tracking daily cortisol concentrations for 2 weeks.

supporting days of measurements without data loss. (Fig. 5E, F, Supplementary Movies 6 and 8). The NFC antenna and WLR were designed and characterized for optimum performance at NFC working frequency of 13.56 MHz (Supplementary Note 8). To mitigate potential antenna detuning caused by changes in the operating environment, the BLOSDA antenna was characterized while attached to the wrist using double-sided medical tape while the WLR antenna was positioned inside a 3D-printed case containing the assembled PCBs, battery, and a ferrite sheet. The Smith charts in Fig. 5D show the corresponding equivalent impedances of the WLR and BLOSDA antennas measured at 13.56 MHz. The impedance characteristics of the BLOSDA NFC tag, tuning capacitor, and antenna demonstrate resonance at 13.56 MHz (Fig. 5G and Supplementary Fig. 19A). The performance of the WLR NFC front-end was assessed by measuring the antenna return loss (S11) where the minimum S11 was −28.56 dB at 13.56 MHz, indicating maximum power transmission at 13.56 MHz (Fig. 5G and Supplementary Fig. 19B). The mechanical stability of the antenna was evaluated by bending the antenna up to 30% to mimic wrist curvature while measuring the return loss (Fig. 5H). To demonstrate the wireless power delivery capabilities of our system, we evaluated the voltage harvested by the BLOSDA NFC tag over various distances between the tag and the reader (either a smartphone or WLR) in Fig. 5I. Furthermore, to confirm the integrity of wireless data communication, we compared the ECG data transmitted by the BLOSDA and received by the WLR in Fig. 5J. This comparison showed no noticable data loss or bit errors, thereby validating the system's reliability for accurate and efficient data transmission. The integration of NFC technology makes the device highly suitable for real-time, and continuous monitoring.

To demonstrate the capabilities of the SQC-SAS wearable in monitoring multimodal signals, controlled experiments were conducted using well-known stressors. The recorded PSI and MSB signals, along with their wavelet power spectrums (WPS) as well as their extracted features, including crest time (CT) from RAP, HR from ECG, and skin conductance level (SCL) and skin conductance responses (SCR) from GSR, are presented in Fig. 5K–M and Supplementary Movie 11. In each experiment, on-body biomolecular and physiological data exhibited significant variations in response to different stressors. These findings emphasize the body's unique reactions to various stress conditions and showcase the SQC-SAS wearable's ability to accurately detect and analyze these changes. During the CPT (Cold Pressor Test) test, cortisol levels increased from a baseline of 86 nM to 104 nM, then gradually decreased to 97 nM by the end of the test. However, cortisol

levels demonstrated a larger change during the math quiz challenge, rising from 86 nM to a peak of 113 nM. HR also varied with the type of stressor, increasing from an average of 89–107 bpm during the quiz, and from an average of 87–95 bpm during exercise. Skin conductance also responded differently across stressors, and ST, although stable during the quiz, showed a slight increase during exercise and a significant decrease during CPT. These results highlight the SQC-SAS wearable's capability to capture physiological variations, enabling continuous monitoring of different types of stress.

On-body assessments were conducted on human subjects to demonstrate the capability of the SQC-SAS wearable for continuous stress monitoring for 24 h. During this study, the subject, wearing the device on the forearm with the WLR positioned for continuous signal recording, engaged in a range of activities including meetings, meditation, eating, running, movie-watching, resting, and sleeping, while both PSIs and MSB levels were monitored. The participant noted the tasks performed throughout the day along with their start and end times, which were then used to annotate the corresponding task periods. For each cortisol level measurement, an MSB restoration process was employed to prepare the sensor for the next reading. This process effectively flushed out the old sweat sample and replaced it with a new one, ensuring that each reading was accurate and reliable. As the results shown in Fig. 6A, in the morning, cortisol levels (MSB signal) peaked at 105 nM, consistent with circadian rhythms, and fell to 79 nM after waking up. However, a check-in meeting elevated cortisol to 95 nM, reflecting cognitive stress, with corresponding increases in HR, frequency of SCR peaks, and SCL. Meditation subsequently reduced cortisol levels smoothly, with also a decrease in HR, SCR peaks amplitude, SCL, and ST, while CT increased. As the subject engaged in physical activities like heavy running, cortisol levels showed a slight increase, rising from 84 to 94 nM, an expected response to physical exertion as the body adapts to increased demands. Exercise also elevated HR from an average of 84–96 bpm, while simultaneously increasing SCR peak incidence, SCL, and ST, with a concurrent decrease in CT. Watching a horror movie significantly elevated cortisol levels to 107 nM, along with increased vascular activity and skin conductivity. In contrast, stable vital signs and cortisol levels were observed during resting. While vital signs remained mostly stable during sleep, a slight decrease in vascular activity and body temperature was evident, with cortisol levels rising gradually in alignment with the body's circadian rhythm (Supplementary Fig. 20). To demonstrate the prolonged capability, we conducted 72 h stress monitoring using the SQC-SAS device, as shown in Supplementary Fig. 21. This extended

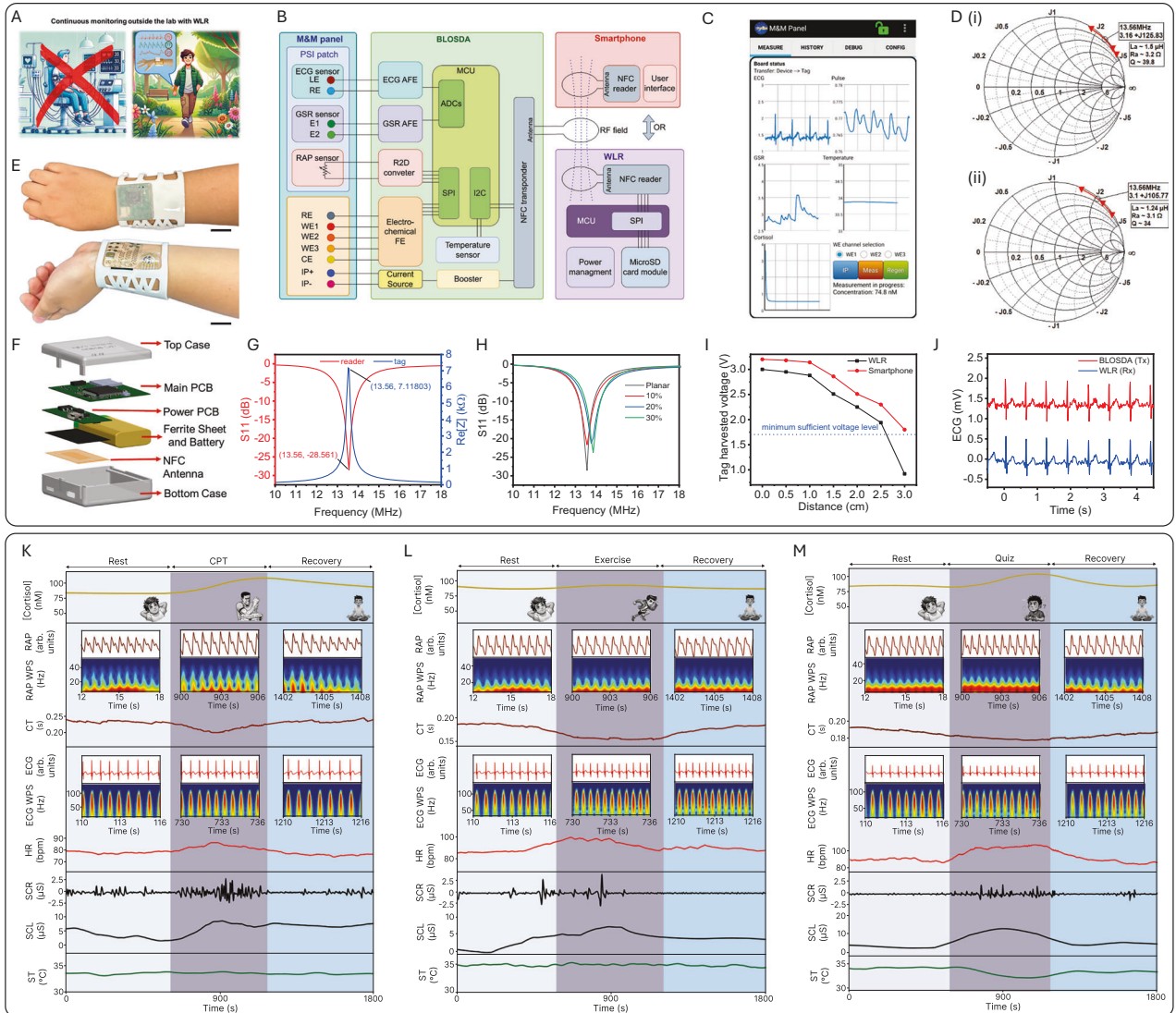

**Fig. 5 | Wireless and continuous multimodal stress monitoring. A** Illustration of wireless data collection using the SQC-SAS wearable. **B** The function block diagram of the SQC-SAS wearable **C** Developed app user interface for the smartphone to communicate with the SQC-SAS wearable. **D** (**i**) WLR and (**ii**) BLOSDA antenna equivalent inductance, resistance, and Q factor measured at 13.56 MHz. The WLR antenna was characterized in the presence of ferrite sheet and PCB components. The BLOSDA antenna was characterized while attached to the human wrist. **E** Optical image of the SQC-SAS wearable wearing on the wrist with the WLR (Scale bar: 2 cm). **F** Structure illustration of the WLR. **G** Measured impedance characteristics of the tuned BLOSDA NFC tag showing resonance at 13.56 MHz. Return loss of the WLR NFC front-end showing the reader delivers maximum power to the antenna at 13.56 MHz. **H** Return loss of the WLR NFC front-end when the antenna is under mechanical bending up to 30%. **I** Voltage harvested by the BLOSDA NFC tag at various distances between the tag and the reader (either a smartphone or WLR), demonstrating adequate wireless power delivery across the tested range. **J** Comparison of ECG data transmitted by the BLOSDA and received by the WLR, confirming the integrity of wireless data communication with no data loss or bit errors detected. Continuous and multimodal monitoring of a subject's stress response and the extracted features (Cortisol level, RAP, RAP WPS, CT, ECG, ECG WPS, HR, SCR, SCL, and ST) under three different stressors: **K** CPT; **L** intense exercise; and **M** quiz.

assessment confirmed the system's operational stability and accuracy in tracking both physiological and biochemical stress markers, thereby demonstrating its robustness for stress profiling.

**Stress detection using machine learning.** The subjective interpretation of physiological signals for stress diagnosis is prone to biases, is time-intensive, and demands expertise from clinicians. In recent years, many researchers have turned their attention to ML and deep neural network (DNN) algorithms. While extracting stress-related features from multiple physiological signals and utilizing them in traditional DNN ML models is a widely adopted approach in stress detection studies[33], deep learning methods offer a fundamental advantage by simultaneously and automatically learning hierarchical representations directly from raw signals. This approach eliminates reliance on predefined features, enabling the model to adaptively extract both local and global patterns across different signal modalities with minimal domain-specific preprocessing. To exploit these advantages, the Inception Multimodal Attention-Based Fusion Deep Neural Network (Inception-MABFDNN) was developed for multimodal feature extraction and stress classification. The architecture integrates a 1D-Inception module (Supplementary Note 10 and Supplementary Fig. 23A), optimized for physiological signal processing. Unlike standard one-dimensional convolutional neural networks (1D-CNN) layers (Supplementary Fig. 23B), which use a fixed kernel size with either max or average pooling, the 1D-Inception module processes inputs in parallel using multiple convolutional kernels ($1 \times 1$, $1 \times 3$, and $1 \times 5$) alongside a parallel max pooling path. This structure captures multi-scale dependencies by analyzing signals at multiple resolutions[68].

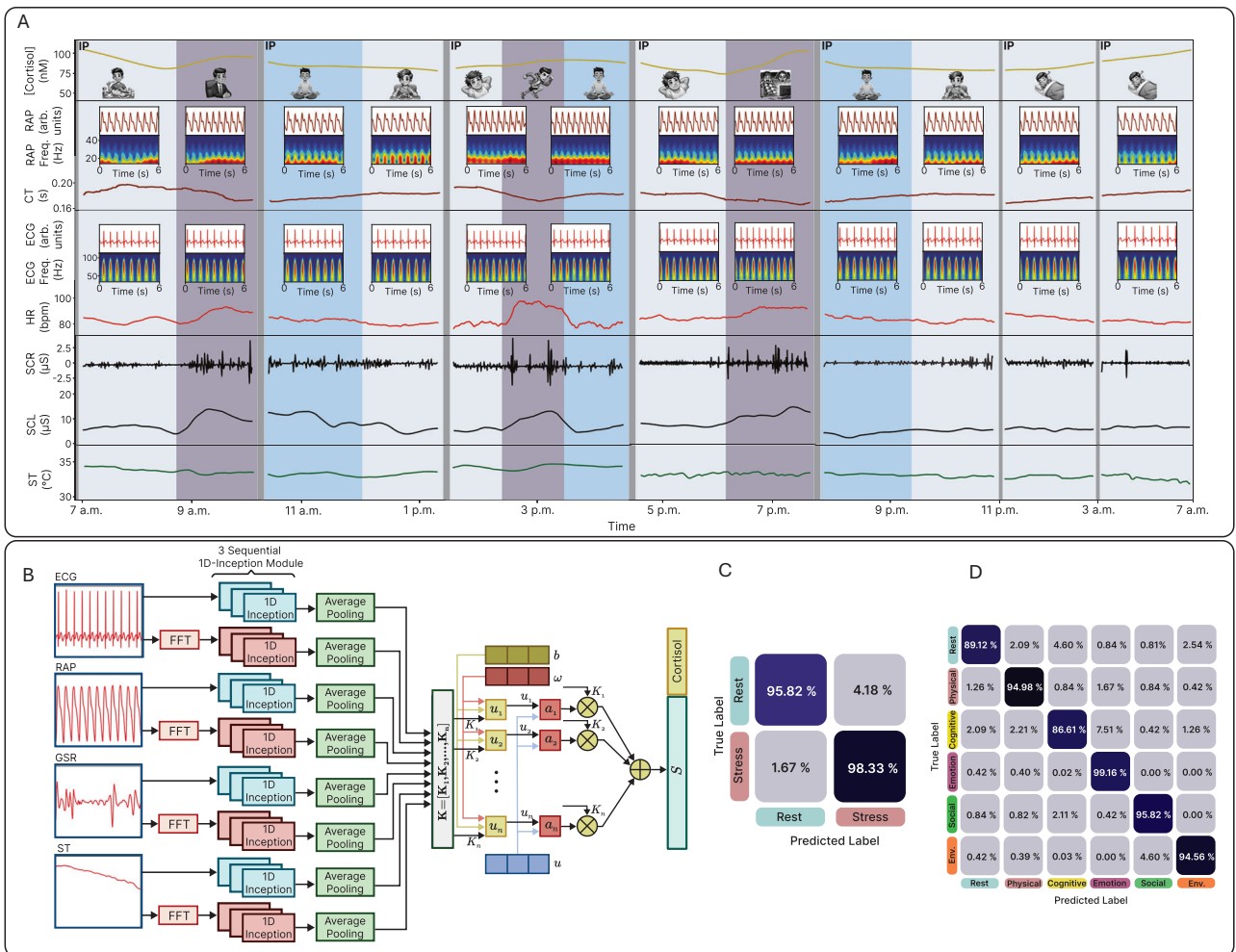

**Fig. 6 | Stress detection using machine learning. A** Wireless and continuous multimodal stress monitoring during various activities. The continuous and multimodal monitoring of a subject's stress response during activities and the extracted features from the data collected by the SQC-SAS wearable (Cortisol level, RAP, RAP WPS, CT, ECG, ECG WPS, HR, SCR, SCL, and ST). **B** Inception-MABFDNN model structure integrating ECG, GSR, RAP, ST, and cortisol signals for stress detection. **C** Representative confusion matrix for subject-independent evaluation in distinguishing stress from non-stress states (labeled as Rest states) and **D** in distinguishing subclasses of stress, including cognitive stress, emotional stress, physical, psychosocial, and environmental stress.

Considering that different patterns extracted from ECG, GSR, RAP, and ST have different effects on predicting stress and its subclasses, we utilized the soft attention (SA) mechanism[69] (Supplementary Note 11) to weigh and fuse all the extracted patterns from all modalities, thus enabling the model to prioritize the most critical patterns for effective stress detection. The Inception-MABFDNN structure is designed to adeptly extract patterns and utilize information from five different signals: ECG, GSR, RAP, ST, and Cortisol (Fig. 6B). This integration of diverse physiological signals into a singular analytical framework contributes to a high detection performance. To ensure a robust evaluation of the model, we employed both subject-independent and subject-dependent approaches. In the subject-independent setting, the model achieved high performance in binary classification of stress versus rest states, with an average accuracy of 93.347% ± 2.103%, and F1-score of 93.323% ± 2.128% (Supplementary Fig. 25A). Under the subject-dependent evaluation, performance increased, yielding an accuracy of 97.027% ± 0.995%, and F1-score of 97.045% ± 0.899% (Supplementary Fig. 26A). For subclassification of stress types, the model attained an accuracy of 90.962% ± 2.815%, and F1-score of 90.878% ± 2.876% in the subject-independent evaluation (Supplementary Fig. 25B). In the subject-dependent case, accuracy was 93.998% ± 2.082%, and F1-score of 93.955% ± 2.107% (Supplementary

Fig. 26B). Representative confusion matrices are shown for the subject-independent classification of stress versus rest (not-stress) states (Fig. 6C) and the subclassification of stress types (Fig. 6D), as well as for the subject-dependent classification of stress versus rest states (Supplementary Fig. 27A) and the subclassification of stress types (Supplementary Fig. 27B). Supplementary Note 12 and Supplementary Fig. 28 contain detailed explanations of our complementary analyses of dataset characterization, the interpretability of stress detection using traditional ML, and the validation of the Inception-MABFDNN design.

A comparative analysis between the developed SQC-SAS device and leading commercial smartwatches is presented in Supplementary Table 3, highlighting critical differences in PSI/MSB sensing capabilities, simultaneous multimodal monitoring, stress detection functionality, and stress subtype analysis. In our study, the performance of the SQC-SAS wearable and commercial smartwatches was systematically evaluated across various conditions, including non-stress states of sitting, standing, and light exercise, and stress states of CPT and math quiz. As illustrated in Fig. 7, commercial smartwatches, which depend solely on a single physiological signal indicator (e.g., PPG), demonstrated limited accuracy and were prone to misclassification errors, including frequent false-positive and false-

negative stress detections. Notably, these commercial devices incorrectly identified neutral activities, such as standing or mild exercise, as stress events, exhibiting accuracies ranging only between 0 and 25% under non-stress conditions. In contrast, our proposed SQC-SAS system, integrating multiple PSI signals and MSB, consistently achieved an impressive ≥90% accuracy rate in distinguishing between stress and non-stress states. Furthermore, commercial smartwatches lack the capability for subclassifying stress. Conversely, the multimodal PSI and MSB sensing, together with continuous, long-term, and simultaneous monitoring features of the SQC-SAS wearable, enable it to not only quantitatively define stress but also differentiate among distinct stress subtypes through comprehensive stress profiling, underscoring the advancement of integrated stress analysis in wearable devices. Supplementary Table 4 provides an additional benchmark comparison between existing wearable sweat biosensing systems reported in the current literature and our SQC-SAS wearable. Our work presents a fully integrated wearable platform that uniquely combines simultaneous, continuous, non-invasive, and autonomous MSB and PSI sensing, enabling quantitative and comprehensive stress assessment and subclassification.

## Discussion

Our Smart Quantitative and Comprehensive Stress Assessor and Sub-Classifier (SQC-SAS) wearable introduces innovative solutions to one of the challenges in health monitoring: the need for continuous, real-time, prolonged, reliable, multimodal, and simultaneous measurement of stress related MSB and PSI signals automatically and precisely. The SQC-SAS wearable is long-term stable, which enables reliable, high-performance operation for extended durations. The SQC-SAS wearable, along with ML, shows the potential for data-driven and quantitative stress assessment and subclassification. This approach facilitates the precise identification and differentiation of distinct subclasses of stress, setting the stage to address the long-standing inconsistencies in stress research. While our ML model demonstrates promising performance in stress sub-classification under the tested conditions, future studies involving larger and more diverse datasets, combined with clinically standardized ground-truth references, will help further support population-level generalization and strengthen the translational relevance of our approach. Additionally, future evaluations of inference latency and model deployment on edge devices can lay the foundation for enabling real-time, on-device stress detection in everyday settings. Also the SQC-SAS wearable incorporates a wirelessly powered IP system, which removes the operational and size constraints of conventional battery-powered IP. This design enables automated, on-demand sweat extraction without user intervention or physical exercise. An integrated flexible microfluidic channel was designed for on-demand sweat sampling and complete and contamination-free sweat sample renewal within a few minutes. These features ensure wireless, autonomous, simultaneous, long-term monitoring of molecular and physiological stress biosignals without the need for a battery. Pairing the device with a WLR designed for efficient power delivery and high-speed data exchange enhances user comfort and enables uninterrupted data collection, extensive data storage, and extended use. In addition, the real-time and on-demand wireless data transmission to a smartphone enables immediate data visualization and supports detailed analysis of stress profiles. With the seamless integration of autonomous multimodal sensing, wireless and battery-free operation, and subclassification of stress, we envision the SQC-SAS wearable as a potential research tool that provides a foundation for unique and precise stress assessment and monitoring for personalized healthcare.

## Methods

### Chemicals and instrumentation

Cortisol, pyrrole, PB, iron chloride, silver flakes, styrene–ethylene butylene–styrene block copolymer (SEBS), sodium chloride, cortisone, progesterone, carbachol, and potassium chloride were procured from Sigma-Aldrich (USA). Toluene was acquired from Fisher Scientific (USA). Sodium phosphate dibasic heptahydrate and sodium dihydrogen phosphate monohydrate were sourced from Thermos Scientific (USA). MAX phase powder was obtained from Nanochemazone (Canada), and PEDOT: PSS (PEDOT, Clevios PH1000) from Heraeus (Germany). Hydrofluoric acid (HF) was obtained from VWR Chemicals (USA) and distilled water (Milli-Q IQ-7000) from Merck (Germany). The gold-coated polyimide film was purchased from KOYE Technology (China). The cotton fabric was acquired from Fabric.com. Medial double-sided adhesive tape (1522), conductive tape (9703), and Tegaderm tape were acquired from 3 M (USA). Transfer tape (Ultra 582U, TransferRite) was acquired from ABI Tape (USA).

All electrochemical cyclic voltammetry and chronoamperometry experiments were conducted using a potentiostat (VersaSTAT 3, AMETEK SI). Impedance measurements were performed using an impedance spectroscope (Zurich Instruments, HF2IS). A linear motor (PS01-23 × 80 R, Linmot) was used for mechanical stability testing. Scanning electron microscopy (SEM) images were captured with a Quanta FEG 250 (FEI Company). The flexible printed circuit board (FPCB) and rigid PCBs were designed in Altium Designer and were fabricated by PCBWay (China). The WLR case was designed in SOLID-WORKS, and 3D printed by Formlabs Form 2 3D printer. The electrodes, microfluidic channels, and screen-printing masks were designed in AutoCAD and were patterned using a cutter plotter (CAMEO 4, Silhouette). COMSOL Multiphysics 6.1 was utilized for FEM simulations to analyze the mechanical stability of electrode designs. Similarly, electromagnetic (EM) simulations for the NFC antenna and antenna matching network were conducted using Ansys Electronics 2022. All experiments related to the characterization of NFC antennas were conducted utilizing a Nano Vector Analyzer. Microcontroller programming for the BLOSDA and WLR was performed in Arduino IDE 2.1 and MCUXpresso IDE 11.4, respectively.

### Fabrication of the PSI patch

**ECG and GSR sensors.** The ECG and GSR electrodes were designed by AutoCAD and made of Au-coated PI film using the developed CPTP methods. The Au-coated PI film was laminated on a transfer tape as a temporary supporting layer for the patterning process. A cutter plotter was used to pattern the desired structure on the film. After the patterning, the excess areas of the Au-coated PI film were peeled off from the transfer tape, and then the remaining electrode structure was transferred to the Tegaderm film. The prepared electrodes with Tegaderm film were kept with the release liner for long-term storage. The electrodes are ready to use and easy to attach like a sticker. The detailed fabrication and assembly procedures are illustrated in Supplementary Figs. 14 and 15.

**RAP sensor.** The RAP sensor was fabricated by developing the conductive fabric, consisting of MXene/PANI/PEDOT-based e-textile, between Tegaderm tape and IDE patterned from Au/PI film. To establish this, the MXene/PANI/PEDOT e-textile was synthesized in accordance with our previous protocol[47]. Initially, MXene ($Ti_3C_2T_x$) nanosheets were synthesized through selective chemical etching[70,71]; 500 mg of MAX phase powder was gradually immersed in aqueous HF and stirred vigorously for 24 h at room temperature. After centrifugation at 3500 rpm for 30 min, the solution was repeatedly washed until it reached a neutral pH. The resulting MXene powder was obtained by drying the solution at 70 °C overnight. The optimized MXene solution (2 mg/mL) was formulated by dispersing MXene powder in distilled water, followed by 10 min of ultrasonication. Subsequently, the MXene/PANI/PEDOT solution was prepared by combining the MXene solution with PEDOT solution and PANI powder (at a 1:1:1 volume ratio). A commercially acquired cotton fabric (100%

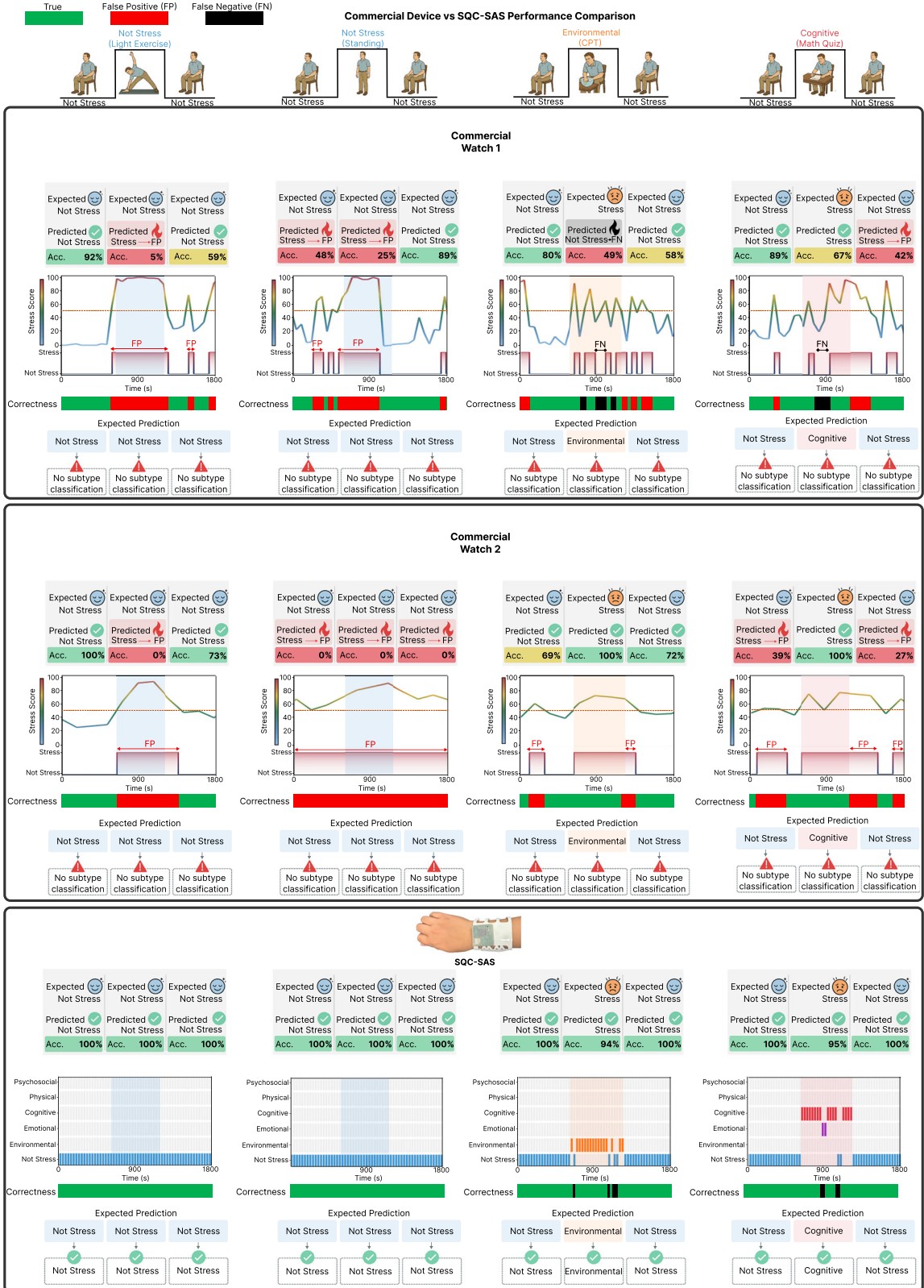

**Fig. 7 | Comparison of stress detection performance between commercial smartwatches and the proposed SQC-SAS wearable under various conditions.** Stress level outputs and prediction accuracies from the Commercial Watch 1 and Commercial Watch 2 are compared against those of the proposed SQC-SAS wearable under multiple scenarios (*n* = 3 technical replicates), including non-stress states (sitting, standing, and light exercise) and stress conditions induced by environmental (CPT) and cognitive (Math quiz) stressors using the subject-independent evaluation approach. Acc. accuracy, FP false positive, FN false negative.

cotton) was immersed in this solution for 3 h and then dried in a vacuum chamber at room temperature.

### Fabrication of the MSB patch
**Fabrication of the SEBS substrate.** Initially, SEBS resin was dissolved in toluene at a ratio of 1:4.35, resulting in the formulation of a highly viscous SEBS mixture. This mixture underwent stirring at 90 °C for a duration of 4 h to attain a consistent and uniform composition. A PET film was employed as a temporary casting substrate. The prepared SEBS resin was subsequently dispensed onto the PET substrate using a doctor blade. Following this, the cast resin was subjected to a curing process lasting 6 h under ambient conditions. The PET substrate was then delicately removed from the resultant SEBS layer, revealing a homogeneous and transparent SEBS film, which was subsequently utilized for subsequent sensor fabrication endeavors.

**Preparation of carbon and silver ink based on SEBS.** The ink formulation consisted of a composite of silver flakes and the previously mentioned SEBS/toluene mixture in a 2:1 ratio. After mixing, the composite was agitated using a vortex mixer for approximately 10 min or until a homogeneous state was achieved. Similarly, the carbon ink formulation was prepared by combining carbon black ink, SEBS, and toluene, followed by vortex mixing for the same duration to ensure uniformity. Before printing, the viscosity of the ink was assessed as a preliminary step. If necessary, incremental additions of toluene (approximately 100 μL/g) were made, followed by further mixing to maintain homogeneity prior to use.

**Electrode printing process.** The prepared SEBS sheet serves as the substrate for the MSB patch printing process. Initially, the upper surface of the SEBS substrate is used for printing interconnections and reference electrodes using silver ink. Subsequently, carbon ink is employed to imprint the working and counter electrodes of the cortisol sensors. After each printing step, the deposited ink patterns undergo a 2-h drying process in a vacuum desiccator. Additionally, for the fabrication of IP electrodes, Au/PI film is precisely patterned using a cutter plotter. These patterned electrodes are then affixed to the lower surface of the SEBS substrate using double-sided adhesive tape.

**Development of the MIP.** The procedure for fabricating the cortisol sensor array based on MIP is illustrated in Fig. 3A. The MIP layer is developed through an electro-polymerization process. Before the MIP deposition, sensor electrodes undergo activation in 1 M $H_2SO_4$, employing CV scans encompassing 20 segments (−1.5 to +1.5 V) at a scan rate of 100 mV/s. The polymerization solution is composed of 0.1 M cortisol template, 5 mM $K_3[Fe(CN)_6]$, 5 mM $FeCl_3$, 0.1 M HCl, and 20 mM pyrrole, combined within a 0.1 M PBS at pH 7.4. After the electro-polymerization procedure, the electrode undergoes triple rinsing with deionized water to eliminate residual chemicals. The implanted template biomolecules within the MIP matrix are then extracted during 20 cycles of over-oxidation via CV, culminating in the creation of complementary cavities. The same preparation methodology is applied to craft the NIP, differing only in the absence of the biomolecule template during polymerization. To maintain consistency with the MIP sensors, the MIP over-oxidation process is conducted, despite the absence of the template within the polymerized layer. The resulting NIP-based electrode is subsequently air-dried at room temperature and subjected to two rounds of cleaning with deionized water.

### Evaluation and characterization of the PSI patch
**Ex-Situ Analysis of the ECG and GSR sensor.** FEM simulation was conducted to analyze the principal strain distribution across three different electrode and interconnection configurations under up to 20% strain along the x-axis, which is larger than the maximum strain tolerated by the skin[11]. Following the FEM simulation, we conducted experiments to verify the stretchability of the three different electrode configurations. All electrodes were first attached to Tegaderm tape for mechanical support. The initial end-to-end resistance of each electrode was first measured by an impedance spectroscope before the mechanical stability tests. The two ends of each electrode were fully clamped to a linear motor to ensure that they were fully subjected to the applied strain. The stretchability test of each electrode configuration was performed by measuring the electrode resistance changes with applied strain until the ribbons were fully ruptured. The electrode configuration that exhibited the highest stretchability was further tested for mechanical stability through a fatigue test involving repeated cyclic bending up to 30% (mimicking wrist curvature[72]) and cyclic stretching up to 20% (covering skin strain tolerance[11]) at a frequency of 0.5 Hz while continuously measuring the resistance of the electrode. Additional mechanical characterization of the PSI panel is provided in the Supplementary Information.

**Ex-Situ analysis of the RAP sensor.** A motorized test stand, equipped with a programmable motor and a force gauge, was used to apply varying pressures to conduct sensitivity and mechanical stability tests. The sensitivity of the RAP sensor was evaluated by monitoring its relative conductance response under applied pressures ranging from 0 to 350 kPa. Additionally, the sensor's response was assessed over five repeated cycles for each pressure level, applying seven different pressure levels within the range of 1–200 kPa. This comprehensive evaluation enabled a thorough assessment of the sensor's sensitivity performance under varying pressure conditions and ensured consistent output under the same applied pressure. The characterization of the conductive fabric used in the RAP sensor, along with the evaluation of the sensor's mechanical stability, is further discussed in the Supplementary Information.

### Evaluation and characterization of the MSB patch
**Preparation of PBS and artificial sweat sample.** A standard solution of 0.1 M PBS at pH 7.4 was prepared, employing sodium phosphate dibasic heptahydrate and sodium dihydrogen phosphate monohydrate salts. To ensure the stability of signal measurements against fluctuations in pH within AS, a buffered solution was employed during the formulation. To this end, 0.1 M PBS (pH 7.4) was combined with the elemental constituents of sweat, encompassing 85 mM NaCl, 13 mM KCl, 17 mM lactate, and 16 mM urea. Furthermore, stock solutions containing known concentrations of cortisol were adeptly devised by amalgamating precise quantities of cortisol with PBS and AS.

**Ex-Situ analysis of the MSB patch.** The comprehensive molecular sensing efficacy of the MSB patch was subjected to scrutiny within a medium comprising 0.1 M PBS (pH 7.4) and AS. ECA was conducted for a duration of 100 s, employing a potential of +0.1 V (versus Ag/AgCl) as the reference electrode. For the calibration plots concerning MIP and NIP-based sensing systems, the cortisol concentration range spanning from 1 nM to 10 μM was scrutinized within both PBS and AS environments. The selectivity assessment was conducted by evaluating the sensor's response to structurally analogous interfering species, specifically cortisone and progesterone. Additionally, the sensor's ability to detect cortisol in the presence of these interfering biomolecules was examined to further validate its specificity.

**Restoration method for the MSB patch.** The restoration process involved applying an optimized voltage of −0.5 V through ECA measurement for optimized seconds, effectively removing the bound biomolecules from the MIP matrix. After the process, the surface properties of the electrodes were assessed by comparing them with a blank concentration current response using CV or ECA techniques. If the current response was lower than the initial blank current, the

subsequent steps were implemented. This involved conducting a CV scan of $K_3[Fe(CN)_6]$ cleaning, which entailed two cycles with a sweep range of −1.2–1.2 V and a scan speed of 0.2 V/s.

**Inducing sweat via iontophoresis stimulation.** For on-demand sweat generation, a muscarinic agonist (carbachol) was employed to promote transdermal sweat production through the IP system. The procedure for sweat stimulation adhered to established methodologies found in earlier literature. The hydrogel containing the muscarinic agent carbachol was synthesized as follows: Initially, a solution comprising 3% w/w agarose was prepared by dissolving agarose in deionized water, followed by continuous stirring and heating to 250 °C until complete dissolution and homogenization were attained. At this point, the solution gradually cooled to 160 °C, whereupon 1% carbachol was incorporated. The resultant mixture was poured into preformed molds resembling electrodes, with gelation transpiring within 10 min under a controlled temperature of 4 °C. The procedure began with cleansing the volunteer's arm using soap and rubbing alcohol, followed by attaching the hydrogel beneath the IP electrodes. The current source on the BLSODA provided a 100 μA current to the IP electrodes, which were wirelessly powered by the WLR or a smartphone via NFC. The current was applied between the IP electrodes for a duration of 5 min. The activation and deactivation of the IP current were controlled through the general-purpose input/output (GPIO) of the microcontroller.

**Assessment of sweat refreshment duration and simulation.** In this study, we conducted an analysis of the refreshment time through rigorous numerical simulations, which were executed using the COMSOL Multiphysics software. To replicate the real-world device accurately, we generated three-dimensional models of various microfluidic designs, maintaining identical dimensions to those of the physical device. These models were constructed utilizing SOLIDWORKS software and subsequently imported into the COMSOL platform. The simulation process encompassed the numerical solution of the Stokes equation, addressing incompressible flow dynamics. This was coupled with the convection-diffusion equation to simulate the mass transport phenomenon.

**NFC antenna characterization.** Subsequent to conducting EM simulations in Ansys Electronics, field measurements were undertaken to assess the performance of the antenna. To facilitate these measurements, SMA connectors were soldered to both the antenna terminals and the NFC reader transmitter terminals. The tag antenna was securely affixed to the wrist using double-sided tape while the reader antenna was placed inside the fully assembled watch case and the equivalent inductances for both antennas were measured. The NFC tag was characterized by using a readout coil connected to a NanoVNA and placing the NFC tag inside the magnetic field of the readout coil to measure the tag's impedance characteristics. Additionally, the NFC reader underwent characterization, specifically directed toward the evaluation of its antenna return loss at the reader's transmitter terminals to assess the maximum power delivered from the NFC reader to the antenna. To further assess the mechanical resilience of the antenna, the two ends of the antenna were securely clamped to a linear motor. After this setup, mechanical stability tests were conducted up to 30% bending while concurrently measuring the antenna return loss. At 30% bending, the resonant frequency shifted to 13.86 MHz, while the S11 at 13.56 MHz was −12 dB which is above the −10 dB threshold which indicates at least 90% of antenna input power is transmitted. This demonstrates the mechanical durability of our fabricated antenna. The effect of read distance on NFC was demonstrated by measuring the voltage harvested by the NFC tag on BLOSDA when using both the WLR and smartphone as the reader. The threshold voltage is 1.7 V required by

the tag for reliable data transmission. The maximum read distance for WLR is 2.5 cm and the smartphone is 3 cm.

## Development of the BLOSDA and WLR

**BLOSDA FPCB.** The readout circuits for all sensors are integrated into a single FPCB. The signals from all the sensors on the panel are measured simultaneously and transmitted to a smartphone via NFC technology. This system utilizes energy harvesting from the NFC antenna, eliminating the need for a battery. The FPCB of BLOSDA integrates multiple components, including a MCU, an NFC transducer (tag) with antenna (NT3H2111, NXP Semiconductors), an ECG AFE circuit (AD8232, Analog Devices), an GSR amplifier (LMP2231, Texas Instruments), a resistance-to-digital (R2D) converter chip (MAX31865, Analog Devices), an electrochemical front-end chip, a current source with boost converter, a multiplexer (MAX4734, Analog Devices), flexible printed circuit (FPC) connectors (Molex), and various passive components. All the chips and passive components are in small packages for compactness. The MCU is programmed with customized firmware to realize data acquisition, data transmission, chip configuration, and power management.

**WLR PCB.** The core of the WLR is an ARM Cortex M3 MCU, an NFC frontend (CLRC663 plus, NXP Semiconductors), antenna, a matching network for the antenna, and a microSD card reader for data storage. It is powered by a 3.7 V LiPo battery, and a voltage regulator chip (TC1264, Microchip Technology) provides a stable 3.3 V regulated voltage to power all the electronics. The battery charging is managed by a charge management controller (MCP73831, Microchip Technology) which effectively regulates the 5 V voltage received from the micro-USB port to 4.2 V for battery charging. The MCU was programmed to use 2 synchronous serial port controllers that use serial peripheral interface communication with the reader chip and microSD card reader for wireless communication and data storage, respectively. To ensure data integrity during NFC communication, the NFC reader and tag were programmed to use a handshake mechanism in which the reader periodically polls the session registers to check if the tag data is available, and the tag polls the session registers to verify if the reader has completed reading the last 4 bytes of SRAM before sending new data. The customized reader firmware was developed in MCUXpresso IDE and was used to program the MCU. The WLR case was designed in SOLIDWORKS, and 3D printed to securely house the PCBs, all electronics, and the antenna.

**Development of a custom mobile application for stress analysis.** The development of the specialized mobile application was accomplished using Android Studio. This mobile application serves as a control terminal for wireless communication between the smartphone and the SQC-SAS wearable via NFC, facilitating the exchange of commands as well as the acquisition, processing, and visualization of multimodal signals. The application boasts a user-friendly interface, empowering users to engage with real-time monitoring and administration of measurement data. Upon the initiation of measurements, users can seamlessly transition to the "measure" tab, which presents dynamic plots depicting the real-time physiological and electrochemical analysis measurements from the PSI and MSB patch. Subsequent to the ECA measurement, the application employs the sensors' calibration equations to translate ECA results into corresponding biomarker concentrations. Furthermore, users can save and export measurement data for further ML-assisted analysis.

## In-situ characterization and validation of stress monitoring utilizing the SQC-SAS wearable
**Ethics.** The validation and assessment of the stress monitoring system were carried out on human subjects, strictly adhering to ethical protocols established and approved by the Institutional Review Board

(IRB; ID: #3062) at the University of California, Irvine. To ensure a comprehensive and diverse participation pool, study participants were recruited from the University of California Irvine campus and surrounding communities. Prior to their involvement in the study, all participants provided explicit written informed consent.

**Cortisol monitoring in accordance with Circadian Rhythms:.** For circadian rhythm experiments, participants were instructed to undertake identical procedures in both the morning (at 10 a.m.) and afternoon (at 5 p.m.) of the same day. The attachment of MIP cortisol sensors to the pre-washed forearms of the subjects was facilitated using double-sided medical tape. Sweat stimulation was achieved through the IP method. The custom Android application was adept at automatically initiating power and data transmission between the sensor panel and the smartphone. The results of the ECA measurements were visualized in real-time through the user interface, and the concentration was computed based on the previously established calibration plot derived from the in vitro experiments.

**Evaluating the influence of stress with the SQC-SAS wearable and dataset collection.** Stress can be categorized into distinct types based on its source and mechanism of action. Cognitive stress arises from mentally demanding tasks or perceived challenges that exceed one's coping capacity, such as problem-solving under pressure. Environmental stress results from adverse external stimuli, including excessive noise, pollution, or sudden temperature changes, that disrupt comfort and physiological regulation. Emotional stress is triggered by intense affective states, particularly negative emotions like fear, anger, or sadness, which elicit autonomic and hormonal responses. Physical stress stems from direct physiological strain, such as intense exercise, injury, or illness, that challenges homeostatic balance. Psychosocial stress emerges from social-evaluative threats and perceived uncontrollability in interpersonal contexts, such as public speaking or social rejection, and strongly activates the HPA axis and sympathetic nervous system[73–75]. To discern and distinguish variations in biosignals between individuals experiencing different categories of stress and those in non-stressful conditions, we employed well-known stress induction protocols specific to each stress type. To induce specific stress categories, we utilized the Stroop test and a time-constrained math quiz for cognitive stress[76], the CPT for environmental stress[77], watching a horror film for emotional stress[78], participation in a simulated interview meeting for psychosocial stress[79], and a high-intensity exercise for physical stress[73]. The online code for the Stroop test was adapted from PsyToolkit, with each question presenting a word on the screen for approximately 500 ms. Participants were required to vocalize the printed color of the word within 1000 ms before moving to the subsequent question. Besides, we administered a math quiz, employing the Big Brain Academy game. The subject was asked to immerse one hand in ice water for the CPT. During the horror movie task, participants were asked to watch a clip from *I Still Know What You Did Last Summer*[80], which has been used in prior related studies[81]. For the intense exercise task, participants ran on a treadmill at their maximum effort for a continuous duration of 10 min[82]. The subjects' forearms were cleaned with alcohol swabs and gauze before affixing the SQC-SAS wearable. A smartphone or WLR was positioned proximate to the antenna on the SQC-SAS wearable's readout circuit to start the recording. To stimulate sweat production, the wireless IP method was employed. Approximately 10 min were allocated for the incubation chamber of the sensor to be filled with sweat. For the stress experiments, subjects initially rested in a quiet room while both physiological and biomolecular signals were recorded for 600 s to establish baseline readings and allow the signals to stabilize. Each subject underwent a pre-stress period of 600 s, referred to as "Rest" or "Not-Stress" period. Following the pre-stress data collection, subjects completed

600 s of each test for each stress category. After each test, the subject was instructed to relax for a 600-s recovery period before the next test. Data was collected from 15 subjects, aligning with the number of participants in the WESAD dataset[83], which serves as a benchmark in deep learning research for stress detection. To account for intra-subject variability in these tests for further ML analysis, each test was performed two times by each subject, where the mental arithmetic task was used in one session and the Stroop test in the other as the cognitive stress test.

**Dataset preparation for machine learning and predicting stress levels.** All the tests to induce different categories of stress are validated stress-inducing protocols based on previous studies; therefore, the duration of these tasks was used to label the corresponding stress states. To enhance model robustness and generalization, data augmentation techniques such as sliding window and jittering are applied[84,85]. In the sliding window approach, the signals were segmented into 5-s windows with a 50% overlap between consecutive windows[86]. For jittering, Gaussian noise with a mean of $\mu = 0$ and a standard deviation of $\sigma = 0.06$ was added to the signals. Also, recent empirical studies have demonstrated that cortisol concentrations in sweat can change within approximately 5 min (300 s) following a stressor[36]. To account for this and the short latency between stressor onset and the initiation of sweat secretion, we applied a 300-s shift to the sweat cortisol signal in our analysis. We evaluated the model using both a fully subject-independent approach and a subject-dependent approach. For subject-independent evaluation, which examines the model's ability to generalize across individuals, we employed Leave-One-Out Cross-Validation (LOOCV) at the subject level. In each iteration, one subject's data was used exclusively for testing. For each held-out subject, models were trained and validated on the remaining subjects, with hyperparameters optimized via grid search using an inner validation split in which one subject was randomly selected for validation and the rest used for training. Having the optimal hyperparameters, the testing procedure was repeated for each subject, and the final performance was reported as the average across all folds. This approach is well-suited for modest sample sizes, as it reduces bias and yields generalization estimates. For subject-dependent evaluation, the model was initially trained on data from all other subjects (excluding the target subject) and then fine-tuned by tuning only the decision layers using data from one session of the target subject. The fine-tuned model was tested on data from the remaining session of the same subject. For visualization of signal variations, HR is extracted from ECG, and CT is extracted from RAP, with both signals smoothed using a 30 s Moving Average (MA) filter. Quadratic interpolation was also performed on the collected cortisol concentration data to better visualize its variations. All signal pre-processing and feature extraction are performed using Python (v.3.9) and SciPy (v.1.15).

**SQC-SAS wearable validation.** To evaluate the performance of the SQC-SAS wearable against commercial systems, each sensor on the device, along with its commercially available counterpart, was affixed to the subject's hands to simultaneously record the corresponding signals. For capturing the ECG and GSR signals, we utilized the MAX30001G evaluation kit with a sampling rate of 128 Hz. Two commercial ECG electrodes were positioned on the subject's left forearm, with a third ECG electrode on the right forearm. Two commercial GSR sensors were placed on the index and middle fingers of the subject's left hand. The commercial PPG signal was captured using the MAX30101 evaluation kit with a 64 Hz sampling rate, with the electrode placed on the subject's left ring finger. ST was recorded using the MAX30205 evaluation kit, with the electrode positioned on the subject's left forearm. The cortisol concentration of the sweat sample was evaluated using the SLV-2930R ELISA kit, and absorbance was measured at 450 nm using the Cytation 5/BioSpa UV-Vis

spectrophotometer. Samsung Galaxy Watch 5 Pro and Garmin Epix Gen 2 smartwatches were used to evaluate stress detection functionality.

The validation of the SQC-SAS wearable's PSI patch against established commercial measurement systems was undertaken employing two statistical approaches. These included the R-squared ($R^2$) score and the Bland-Altman plot. The $R^2$ score was calculated using the formula: $R^2 = (\frac{\sum (x_i - \bar{x})(y_i - \bar{y})}{\sqrt{\sum (x_i - \bar{x})^2 \sum (y_i - \bar{y})^2}})^2$. In this equation, "$x_i$" represents either the extracted feature values from 60-s intervals of physiological signals or the cortisol concentration measured by the commercial system, "$\bar{x}$" signifies the mean of all "$x_i$" values, "$y_i$" corresponds to either the extracted feature values from 60-s intervals of physiological signals or the cortisol concentration measured by the SQC-SAS wearable, and "$\bar{x}$" denotes the mean of all "$y_i$" values. This calculation aims to illustrate the linear correlation between the concentrations measured by commercial systems and those obtained using the SQC-SAS wearable. Additionally, the Bland-Altman plot, another statistical technique, was employed to delineate the disparities between the concentrations determined by commercial systems and the corresponding measurements acquired through the SQC-SAS wearable. The confidence intervals for the Bland-Altman Limits of Agreement were calculated at 1.96 times ""$\sigma_D$" where "$\sigma_D$" is determined by the formula: $\sigma_D = \sqrt{\frac{D_i - \bar{D}}{N-1}}$. In this formula, "$D_i$" signifies each individual variance between the two sets of measurements, "$\bar{D}$" represents the average of all "$D_i$" values for the specific value of interest (extracted feature values or measured cortisol concentration), and "$N$" is the number of measured samples. All statistical analyses were conducted using Origin Pro software.

## Reporting summary

Further information on research design is available in the Nature Portfolio Reporting Summary linked to this article.

## Data availability

All data supporting the findings of this study are available within the article and its supplementary files. Request for any additional information can be directed to the corresponding authors and will be fulfilled upon reasonable request. Source data are provided with this paper.

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

## Acknowledgements

This work was supported in part by start-up funds provided to R.E. by the Henry Samueli School of Engineering and the Department of Electrical Engineering and Computer Science at the University of California, Irvine, and in part by the National Science Foundation (Award 2538997 to R.E.).

## Author contributions

R.E.—conceived the idea, designed research and experiments, and supervised the study; X.P, A.G, S.C, and J.R.—performed experiments with assistance from J.A.T.; X.P, A.G, S.C, and J.R.—performed the formal analysis, investigations, and visualizations; X.P, A.G, S.C, J.R, J.A.T., and R.E.—drafted the paper; all authors provided feedback.

## Competing interests

The authors declare no competing interests.
