## [Transparent Peer Review file · Nature Communications]

A Quantitative, Multimodal Wearable Bioelectronic for Comprehensive Stress Assessment and Sub-Classification

Corresponding Author: Professor Rahim Esfandyarpour

Version 1:

Reviewer comments:

Reviewer #1

(Remarks to the Author)

This work demonstrated an impressive multimodal wearable bioelectronic device integrating biophysical signals such as ECG, GSR, and RAP with sweat cortisol molecule detection for stress monitoring. The system is highly comprehensive with abundant investigations and detailed characterizations, achieving daily, continuous, and high-precision stress assessment and subclassification aided by wearable data acquisition and machine learning algorithms. The authors also verified the performance in real scenarios and the superiority over commercial smart watches. The work is of interest to the field, despite some issues that need to be further supplemented and clarified.

1. Fig.1 contains too much information and abbreviations to understand clearly. It is recommended to adjust the figures, e.g., move Fig 1G to 1B for the clarification of PSI and MSB. Second, Fig. 1B and 1C seems a little duplicated with Fig. 1A scheme, it may be optimized or some schemes may be moved to SI.
2. Fig.2, the authors demonstrated electrode stability over bending and stretching cycles. Since the resistance still changed up to 2 folds, the authors may provide analysis in how much change in the resistance would allow stable signals in principle.
3. In Fig. 2I, there is an ambiguity between the number of electrodes marked in the figure and in the figure caption. Is it marked in reverse?
4. It is recommended to move the characterization of E-textile materials (Fig 2O and 2P) to SI, or display an SEM image showing the fiber morphology clearly.
5. Fig.3A, it is recommended to add the molecular structure of the used MIP, i.e., PPy. Also it is suggested to move Fig.3Q and 3T to SI, and remove Fig.3R(i). In addition, for all the current response and recovery data in Fig.3, it is recommended to indicate the cortisol concentration in the measurements, similar to Fig.3T.
6. Fig.3 caption, the last sentence was repeated twice.
7. The main concern is the regeneration of the MIP biosensor. First, it should demonstrate at least 10 cycles of regeneration, as recent literature showed (doi.org/10.1016/j.scib.2025.03.060). Second, it would be helpful to provide further explanation or assumptions as to why ECA outperforms other methods in terms of regeneration time. Third, how did the regeneration voltage as well as the sweat refreshing process affect the reuse? Finally, the MIP regeneration mechanism seems impractical to reflect real-time concentration fluctuation, therefore the cortisol results in Fig. 5 and Fig. 6 are confusing to exhibit continuous level changes. Previous literatures (e.g., *Sci. Adv.* 2022, 8, eabk0967) usually report isolated cortisol level points with a certain time interval, which is more reasonable. The author can provide how many cortisol measurements were continuously conducted in Fig. 5 and Fig. 6.
8. The author can clarify more clearly for the subclasses of stress to maintain consistency for Fig.6A, 6H and Fig.7.
9. The Inception-MABFDNN model mentioned in the article performs well in stress classification, but the interpretability of the model is insufficient. It is suggested to add detailed explanations of the model signal preprocessing, feature extraction and decision-making process in order to better understand its performance under different stress states.
10. The sample size distribution of the stress dataset is not disclosed, which may introduce classification bias. Also the inference delay and edge deployment feasibility of the Inception-MABFDNN model have not been evaluated. Can the authors consider providing the category distribution statistics of the dataset and explain the adopted sampling/processing strategy and its impact on the robustness of the model?
11. The title claimed "prolonged stress assessment", but it only means "long-term storage of stress profiles", and the monitoring is still limited to 24 h, which is a bit misleading. The title and abstract needs refinement to point out the novelty of

this work.

Reviewer #2

(Remarks to the Author)

The authors presented wearable electronics with machine learning for prolonged stress assessment. The topic itself is quite interesting and timely, but the authors overemphasizes sensor integration, trying to claim as many sensors as possible at the expense of depth in sensor validation and human data. In addition, the wearable sensor results reported in the paper are inconsistent with previous studies. For example, sweat cortisol concentration is typically a few nM, yet the authors reported several hundreds of nM concentrations, 100X than established studies. The authors showed 4 data points in their validation results. A detailed result validation is needed to justify the authors' findings. It is therefore not recommended for publication unless significant justification is presented. Some additional concerns are listed below:

1. The authors integrated radial artery pulse, galvanic skin response, skin temperature, MIP cortisol, and ECG. With so many sensors integrated together, power supply is a big issue. It is crucial to consider batteries and battery life in a complete wearable system to support the proposed prolonged measurement.
2. The authors claimed the major significance they achieved was prolonged stress state classification for the first time. But such classification has been reported (Nature Electronics, 7, 168-179, 2024), with extended stress classification and assessment. In addition, the authors appeared to involve only acute stressors, failing to justify the "prolonged" period as proposed in the title and abstract.
3. The cortisol sensor shown in Fig3 needs more than 100 s to stabilize, and keeps drifting during the time window. If cortisol sensor cannot be stabilized, the on-body data validity is questionable. In addition, the authors provided calibration plot of 1 nM to 1 μ M with wide variations, which is well above the physiological range. Cortisol concentrations in human sweat is known less than 10nM, and the MIP cortisol sensor in this paper is not meeting the requirements.
4. The cortisol sensor claimed in the paper also showed much inconsistent results with previous studies (Matter 2, 4, 921-937, 2020; Science Advances, 8, eabk0967, 2022; Scientific Reports 10, 19050, 2020). Sweat cortisol should be in the range of ng/mL, yet the authors report 100X higher result. Such high concentrations was known to exist only in serum not in sweat. The ELISA validation in Fig4 is also questionable, with only 4 data points collected. A detailed result validation is needed to justify the authors' findings.
5. The authors claimed 24 hour of continuous cortisol monitoring, but the long-term stability of cortisol sensor, as well as other sensors were not tested in vitro. Detailed sensor validations are needed before human studies.
6. The authors designed 2-lead ECG sensors together with GSR sensor. Without an additional reference lead, the potential difference can be easily interfered with environmental noise, especially with the applied potential for GSR. GSR sensor is an impedance-based sensor and require a voltage input, how could the 2-lead ECG data collected without suppressing common-mode interference? The ECG baseline in Fig5 seems totally isolated from GSR signals, which cannot be achieved with the design in the paper.
7. The authors claimed machine learning for analyzing stress assessment. Machine learning combining physical and chemical parameters is an important direction for mental health quantification. However, only 3 subjects' data is collected. The dataset itself is insufficient to perform any serious study on biomarker monitoring. Analyzing such few data cannot separate training dataset from testing dataset, and will cause model overfitting.

Reviewer #3

(Remarks to the Author)

Review of "A Quantitative, ML-Powered, Multimodal, On-Site Regenerative Wearable Bioelectronic for Prolonged Stress Assessment and Sub-Classification in Daily-Life Contexts"

This manuscript presents the development and evaluation of the Smart Quantitative and Comprehensive Stress Assessor and Sub-Classifier (SQC-SAS), a wearable bioelectronic patch designed for long-term, objective, multimodal stress monitoring and subclassification in real-life environments. The patch integrates molecular and biophysical sensors, autonomous wireless iontophoresis for sweat stimulation, and machine learning (ML) for stress detection and classification. The authors claim high accuracy in both stress detection (>95%) and sub-classification, and position the device as a clinically actionable tool for stress research and personalized health interventions.

The manuscript addresses a highly relevant challenge in stress research by aiming to develop an objective, continuous, and real-world applicable tool for stress quantification. The technical aspects, such as microfluidic integration, physiological sensing, sensor regeneration, and wireless operation, are undoubtedly impressive and represent a substantial engineering advancement.

However, I believe the claims regarding the device's capability to assess and sub-classify stress in real-life situations are currently overstated. Given the small number of participants, limited number of in-situ data points (e.g., only four for cortisol correlation analysis), and lack of robust objective reference measures, the presented findings should be interpreted more cautiously. The ML-based classification results, in particular, risk being overfitted and not generalizable. I would recommend framing the in-situ and ML components as promising *proof-of-concept* results rather than *validated conclusions*. In my opinion, this would not diminish the importance of the work, which already marks a significant step forward, but would more accurately reflect the current level of validation. On that note, the authors should also consider reframing the discussion towards the real-world applicability of the device, including its potential for future clinical validation, but without overstating the current findings and clearly acknowledging the limitations.

In summary, I recommend major revision. The authors are encouraged to reframe their claims more cautiously and provide a clearer, more critical discussion of the limitations. A reframing of the study as a proof-of-concept investigation, rather than a fully validated clinical tool, would significantly enhance the credibility and clarity of the work. A more thorough treatment of methods, results, and limitations is essential to improve the manuscript's robustness. Below, I provide detailed comments.

Introduction:

- * The introduction provides a broad overview of the health implications of stress and the need for improved stress monitoring. However, the claims about the current state-of-the-art and the novelty of the presented device are somewhat overstated and would benefit from clearer contextualization within existing literature.
- * The term "biophysical stress indicators" is used throughout the manuscript but is uncommon in the psychophysiology literature. Consider replacing this term with "physiological" or clearly defining it early on.
- * Some claims regarding the impact of stress on mental and physical health lack appropriate and recent references. For instance, the claim in ll. 50–52 is not supported by a reference. The paper reference before and after this sentence is from 2008, and the claims are made for 2022, so the reference cannot be the same. Please provide a correct reference for this statement.

Methods:

- * The number of participants involved in the study is not clearly stated. It appears that only three participants were included. This extremely small sample size limits generalizability and significantly impacts the robustness of the ML analysis. The authors should justify this sample size and discuss its limitations explicitly. Moreover, the demographic information (age, gender, health status) of participants should be clearly presented.
- * Section: "Dataset Preparation for Machine Learning and Predicting Stress Levels": Why is the Stroop test not included in the analysis, even though it was mentioned in the preceding section?
- * It is stated that the cortisol signal was shifted by 300 seconds to align with physiological signals due to the natural sweat delay. However, it is known that cortisol (if measured in saliva) is released within 10-15 minutes *after* the stressor. This should be taken into account when interpreting the results and the authors should clarify this point and the temporal association between salivary-based cortisol and sweat-based cortisol.
- * The cortisol samples collected at defined times of day (10 am and 5pm) can only partially reflect the circadian rhythm of cortisol, especially if no other important factors (awakening time, physical activity, etc.) were controlled for. Also, it is not clear from the manuscript whether reference cortisol samples (e.g., from saliva) were taken. It is okay to only measure cortisol at two time points for a proof-of-concept study, but the claims to "measure circadian rhythms" are somewhat overstated.
- * Which reference measures were used to assess the participants' stress levels in the laboratory study (Stroop test, etc.) or was the "task" itself used as a reference label? This should be stated more clearly.
- * How were stress periods annotated during the 24-hour monitoring part of the study?
- * Details on reference measures for stress validation (e.g., self-report, salivary cortisol, behavioral tasks) are insufficient. Were salivary cortisol samples collected? If so, how were they analyzed and aligned with the sweat-based data?

Results:

- * How were the machine learning models evaluated? Was a cross-validation performed or was the dataset split into training and test sets? This should be clarified in the text. Given the small sample size, but the large complexity of the ML model, it is likely that the model is overfitting to the training data if no adequate counter-measures were taken. This can also explain the high accuracy of the model, which can likely be due to overfitting.
- * Figures 4O and 4P report correlation and Bland-Altman plots for cortisol validation based on only four data points. This is insufficient to draw meaningful conclusions. Please reframe this analysis as a preliminary proof-of-concept rather than validation.
- * How were the different stress labels defined? This is somewhere mentioned in the text, but I would suggest to provide a more structured overview of the different stress labels and how they were defined (also complemented with the corresponding literature references).
- * Fig. 7: The comparison between the commercially available devices and the wearable patch is biased towards the wearable patch. If the data recorded in the study is used to train the ML model, it is not surprising that the wearable patch performs significantly better than the commercially available devices, especially given the small sample size. It is also not surprising that the patch is reliably capable of detecting different subtypes of stress if the very same data is used to train the model and the model was explicitly trained to detect these subtypes.

Discussion:

- * Given the length of the paper, the discussion section is very short and no limitations of the study are mentioned at all. Major limitations, including small sample size, lack of independent reference standards, limited validation, and potential overfitting, should be clearly discussed. Explicitly stating the limitations of the study, and outlining the next steps needed to improve generalizability and clinical validity (e.g., larger sample, external validation, longer monitoring) would significantly enhance the manuscript's robustness.

Minor Comments:

- * Please provide the IRB approval number.
- * The use of abbreviations is excessive and impairs readability. Consider reducing abbreviations or including a glossary.

- * A reference for the Bland-Altman method should be provided (e.g., Altman & Bland, 1983).
- * "Reduce stress markers" (p. X) should be reworded to "reduce stress marker levels" or similar.

Version 2:

Reviewer comments:

Reviewer #1

(Remarks to the Author)

The authors have addressed the reviewers' concerns well, although still lots of figures are remained in the maintext, which I would suggest to move them to SI, such as Fig.2O-Q, Fig.3R(i). In addition, the authors stated continuous monitoring of cortisol with a 3-minute interval, thus I still suggest to present the cortisol concentration data in Fig.5K-M as isolated dots instead of continuous lines for better illustration, since the total monitoring time was only 30 min for 10 data points.

Reviewer #2

(Remarks to the Author)

The authors have addressed my previous concerns. I would like to recommend its publication.

Reviewer #3

(Remarks to the Author)

I would like to thank the authors for revising their manuscript and addressing my previous comments. The revised version is much improved, and I appreciate the authors' efforts to clarify various aspects of their work. However, I still have some concerns regarding the machine learning (ML) methodology and the presentation of results that should be addressed before the manuscript can be considered for publication.

The presentation of the ML methodology still lacks sufficient detail. While the authors have added some more information, it still does not allow to fully understand the approach taken. Most importantly, in the caption of Fig. 6, an XGBoost classifier is mentioned based on which feature importances were computed (Fig. 6E). However, this is the only mention of XGBoost in the entire manuscript. It is unclear why this classifier was chosen, how it was trained, and how its performance compares to the DL-based model (Inception-MABFDNN) for which I assume the rest of the results in Fig. 6 were obtained. Furthermore, the rest of this section ("Machine Learning Powered Stress Detection") is still quite vague and lacks clear structure. The authors write about feature extraction and the "classical machine learning pipeline" (Fig. 6B), but in the end they seem to have used a deep learning model. It is unclear how these two components relate to each other. Similarly, the chord diagram in Fig. 6C is mentioned briefly, but without really explaining its purpose and the insights it provides or implications for the rest of the analysis.

Reviewer #4

(Remarks to the Author)

This manuscript describes a technically sophisticated wearable platform that integrates molecular and physiological sensing modalities with machine learning-driven analysis for stress assessment. The work is ambitious and comprehensive, with significant engineering achievements. Its principal innovation lies in the integration of multiple sensing strategies into a single, regenerable, multimodal system, rather than in the novelty of any individual sensing element.

Comments:

- The technical complexity of the system is substantial. Each individual sensor is state-of-the-art.
- The study is best regarded as a proof of principle. The limited number of human participants and restricted machine learning training data constrain the robustness and generalizability of the findings. Nevertheless, the feasibility results highlight the value of multimodal sensing when coupled with machine learning models.
- The comparison against commercial wearables is one of the most compelling parts of the study. It demonstrates that combining multimodal molecular and physiological data streams enables more accurate classification of stress states than conventional approaches.
- The device architecture, comprising soft sensors, flexible substrates, in situ regeneration, wireless iontophoresis, and a coupled readout module, is technically interesting. However, for translation, the system may be cumbersome. A simplified design, such as a battery-powered flexible module paired with replaceable sensor patches, could represent a more practical path toward long-term, real-world adoption.
- The manuscript relies heavily on acronyms, which reduces clarity even with reference tables. This density makes it challenging to follow the contributions of individual components. As an example deciphering the relative role of cortisol in the machine learning model from the figure is difficult.
- The discussion occasionally overclaims novelty. Several statements of "first-time" demonstrations, particularly regarding iontophoresis in wearable systems, are not entirely accurate. The data and technical integration are strong enough to stand without such claims, and moderation in tone would strengthen the manuscript.
- Overall, the manuscript is comprehensive, with detailed descriptions of device design, validation, and performance. The work demonstrates significant technical capability and adds valuable evidence for the potential of multimodal, ML-powered stress monitoring.

Summary:

This is a rigorous and ambitious manuscript that makes a meaningful contribution to the field of wearable bioelectronics. Its

innovation lies in the system-level integration of regenerable molecular sensing and physiological monitoring into a single platform for stress assessment. While the work remains a proof of concept with limited generalizability at present, it provides a strong foundation for future large-scale validation.

Version 3:

Reviewer comments:

Reviewer #3

(Remarks to the Author)

my concerns were sufficiently addressed

Reviewer #1:

This work demonstrated an impressive multimodal wearable bioelectronic device integrating biophysical signals such as ECG, GSR, and RAP with sweat cortisol molecule detection for stress monitoring. The system is highly comprehensive with abundant investigations and detailed characterizations, achieving daily, continuous, and high-precision stress assessment and subclassification aided by wearable data acquisition and machine learning algorithms. The authors also verified the performance in real scenarios and the superiority over commercial smart watches. The work is of interest to the field, despite some issues that need to be further supplemented and clarified.

Author Response: We sincerely appreciate the reviewer's thoughtful and encouraging comments regarding our work. We are grateful for your recognition of the comprehensive design, detailed characterization, and practical validation of our multimodal wearable bioelectronic system. Your positive feedback is truly motivating, and we highly value your insightful suggestions, which have provided us with an excellent opportunity to further strengthen and clarify our manuscript.

All changes are marked in “**Blue**” in the revised manuscript for the reviewer's convenience.

Thank you again for your valuable feedback.

1. Fig.1 contains too much information and abbreviations to understand clearly. It is recommended to adjust the figures, e.g., move Fig 1G to 1B for the clarification of PSI and MSB. Second, Fig. 1B and 1C seems a little duplicated with Fig. 1A scheme, it may be optimized or some schemes may be moved to SI.

Author Response: We sincerely appreciate the reviewer's thoughtful suggestion regarding the organization and clarity of Fig. 1, which indeed prompted us to carefully reevaluate the figure layout and its intended message. Our aim with this figure was to create a logical visual flow that aligns with the system development and operational workflow of the SQC-SAS wearable. Specifically, Figure 1A introduces the overall system concept, followed by Fig. 1B, which presents the layer-by-layer assembly of the complete SQC-SAS device to highlight its integrated construction. Figure 1C then focuses on the individual PSI sensors, showcasing the different biophysical sensing modalities (ECG, GSR, RAP), while Fig. 1D provides a detailed visualization of the MSB component, emphasizing the sweat induction mechanism, electrode arrangement, and iontophoresis process. Following this fabrication-oriented sequence, Fig. 1G was purposefully placed at the end to underscore the real-world, long-term application of the system in continuous stress monitoring across everyday settings. We fully agree with the reviewer on the importance of clarity and logical flow, and we have taken this opportunity to further refine the figure caption to enhance readability and better guide the reader through the figure's content. We are grateful for the reviewer's constructive feedback, which helped us strengthen the presentation and accessibility of our work.

(Please check Manuscript Page No. 4-5)

2. Fig.2, the authors demonstrated electrode stability over bending and stretching cycles. Since the resistance still changed up to 2 folds, the authors may provide analysis in how much change in the resistance would allow stable signals in principle.

Author response: We sincerely thank the reviewer for this insightful comment. We would like to clarify that the resistance change observed in **Fig. 2E** is approximately 2%, not a two-fold increase. This minor variation reflects the inherent mechanical compliance of the electrode under repeated bending and stretching. Importantly, as illustrated in **Fig. 2J**, the electrode resistance R_{AU} ($\sim 1 \Omega$) is substantially lower than the skin-electrode contact resistance ($\approx 10.6 \text{ k}\Omega$). As a result, even when subjected to mechanical deformation, the slight fluctuation in electrode resistance exerts a negligible effect on the overall electrode-

skin impedance, as confirmed by the stable measurements shown in Fig. 2K. This negligible impact ensures consistent signal acquisition and highlights the robustness of our electrode design in maintaining reliable performance under dynamic conditions. To enhance clarity for the readers, we have also revised the corresponding explanation in the manuscript. We are grateful to the reviewer for providing this valuable feedback, which allowed us to further strengthen the presentation of our results.

(Please check Manuscript Page No. 7-8)

3. In Fig. 2I, there is an ambiguity between the number of electrodes marked in the figure and in the figure caption. Is it marked in reverse?

Author response: We sincerely thank the reviewer for carefully pointing out this ambiguity regarding the labeling of the electrodes in **Fig. 2I**. We apologize for the oversight and any confusion it may have caused. In response, we have carefully revised the figure caption to ensure consistency between the electrode markings in the figure and the description in the caption. We appreciate the reviewer's attentive feedback, which helped us improve the clarity and accuracy of the presentation.

(Please check Manuscript Page No. 8)

4. It is recommended to move the characterization of E-textile materials (Fig 2O and 2P) to SI, or display an SEM image showing the fiber morphology clearly.

Author Response: We sincerely acknowledge the reviewer's valuable comment. In accordance with the suggestion, we have moved the characterization of the e-textile materials (**Fig. 2O-P**) to the Supporting Information and have revised the manuscript accordingly. We thank the reviewer for this helpful recommendation.

(Please check Supplementary Note. 3, Supplementary Fig. 4 and Supplementary Page No. 5-6 & 24)

5. Fig. 3A, it is recommended to add the molecular structure of the used MIP, i.e., PPy. Also it is suggested to move Fig. 3Q and 3T to SI, and remove Fig.3R(i). In addition, for all the current response and recovery data in Fig.3, it is recommended to indicate the cortisol concentration in the measurements, similar to Fig.3T.

Author Response: We sincerely thank the reviewer for the valuable suggestions on improving the clarity and presentation of **Fig. 3**. In response to the comments, we have added the cortisol concentrations to all current response and recovery plots to ensure consistency and facilitate interpretation. We have also included the molecular structure of the molecularly imprinted polymer (PPy) in the figure caption for better clarity. Additionally, as per the reviewer's recommendation, we have moved the relevant detailed information to the Supporting Information and revised the manuscript accordingly. We greatly appreciate the reviewer's insightful feedback, which helped us enhance both the quality and clarity of our work.

(Please check Supplementary Note. 4, Supplementary Fig. 11 and Supplementary Page No. 6 & 31)

6. Fig. 3 caption, the last sentence was repeated twice.

Author Response: We sincerely thank the reviewer for carefully pointing out the repetition in the last sentence of the Figure 3 caption. We apologize for this oversight and have corrected it in the revised manuscript. We appreciate the reviewer's attention to detail, which helped us improve the clarity and quality of the presentation. *(Please check Manuscript Page No. 11)*

7. The main concern is the regeneration of the MIP biosensor. First, it should demonstrate at least 10 cycles of regeneration, as recent literature showed (doi.org/10.1016/j.scib.2025.03.060). Second, it would be helpful to provide further explanation or assumptions as to why ECA outperforms other methods in terms of regeneration time. Third, how did the regeneration voltage as well as the sweat refreshing process affect the reuse? Finally, the MIP regeneration mechanism seems impractical to reflect real-time concentration fluctuation, therefore the cortisol results in Fig. 5 and Fig. 6 are confusing to exhibit continuous level

changes. Previous literatures (e.g., Sci. Adv. 2022, 8, eabk0967) usually report isolated cortisol level points with a certain time interval, which is more reasonable. The author can provide how many cortisol measurements were continuously conducted in Fig. 5 and Fig. 6.

Author Response: We sincerely thank the reviewer for the thoughtful and constructive comments regarding the regeneration of the MIP biosensor. In response, we have thoroughly optimized and validated the sensor's reusability by demonstrating ten consecutive regeneration cycles, which confirmed stable performance and reliable signal output (*please check Fig. 3H-I* in the revised manuscript). To clarify the superior regeneration efficiency of ECA compared with other methods such as DPV and CV, we emphasize that ECA enables the application of a constant potential during regeneration, which effectively removes the negatively charged cortisol molecules from the MIP surface via electrostatic repulsion. Unlike techniques using fluctuating potentials, ECA minimizes the risk of generating unwanted functional groups on the polymer surface, which is critical for preserving the selectivity of the MIP sensor. We conducted systematic optimization studies by varying the applied voltage and regeneration time, ultimately selecting -0.5 V for 55 seconds as the optimal condition to achieve efficient regeneration without compromising sensor performance (*Please check Manuscript Page No. 12 & 26*). Furthermore, we recognize that both the regeneration voltage and the sweat refreshing process can influence sensor reuse and signal stability. To address this, we carefully optimized the microfluidic design using COMSOL simulations (*please check Fig. 3V and Supplementary Fig. 12*), allowing controlled sweat flow with a four-inlet configuration that ensures effective analyte clearance and prevents performance degradation. Regarding the continuous cortisol monitoring presented in Figures 5 and 6, each measurement was performed at 3-minute intervals, comprising approximately 100 seconds of sensing followed by 55 seconds of regeneration, which allowed us to reliably track dynamic cortisol fluctuations over time. Additionally, to support this protocol, we have included a demonstration of the both ex-situ and in-situ regeneration process in the Supporting Video (*Supplementary Movies 9 and 10*), highlighting the sensor's capability for repeated, real-time measurements. All these points have been clearly addressed and elaborated in the revised manuscript. We are grateful for the reviewer's insightful feedback, which helped us improve the clarity and scientific rigor of our work.

8. The author can clarify more clearly for the subclasses of stress to maintain consistency for Fig.6A, 6H and Fig.7.

Author Response: We sincerely appreciate the reviewer's kind and valuable comment, which has helped us improve the clarity and consistency of our manuscript. In response, we have provided a more detailed explanation of the defined stress subclasses along with the corresponding induction protocols for each stress type, ensuring that the categorization is clearly conveyed. Furthermore, we have carefully revised Fig. 6 and 7 to maintain consistency in the naming and presentation of stress subclasses throughout the manuscript. We are grateful to the reviewer for this insightful suggestion, which allowed us to enhance both the scientific clarity and readability of our work.

(*Please check Manuscript Page No. 28-30 and Fig. 6 & 7*)

9. The Inception-MABFDNN model mentioned in the article performs well in stress classification, but the interpretability of the model is insufficient. It is suggested to add detailed explanations of the model signal preprocessing, feature extraction and decision-making process in order to better understand its performance under different stress states.

Author Response: We sincerely thank the reviewer for this insightful and valuable comment, which highlights an important aspect of our study. We apologize for any misunderstanding caused by the initial presentation. To address this, we would like to clarify that we have provided a comprehensive description

of the signal preprocessing steps, feature extraction methodology, and the decision-making process of the Inception-MABFDNN model in the Supplementary Note. This includes a transparent explanation of the preprocessing pipeline, the model's approach for extracting multi-resolution features from physiological signals, and the design and rationale of the attention-based decision-making module. We truly appreciate the reviewer's thoughtful feedback, which allowed us to ensure greater clarity and completeness in presenting our machine learning framework.

*(Please check **Supplementary Notes 9, 10 & 11** and **Supplementary Fig. 21-22**)*

10. The sample size distribution of the stress dataset is not disclosed, which may introduce classification bias. Also the inference delay and edge deployment feasibility of the Inception-MABFDNN model have not been evaluated. Can the authors consider providing the category distribution statistics of the dataset and explain the adopted sampling/processing strategy and its impact on the robustness of the model?

Author Response: We sincerely thank the reviewer for this highly valuable and insightful comment, which addresses critical aspects of dataset transparency and model applicability. In response, we have added a detailed description of the dataset's category distribution and the adopted data processing and sampling strategies in the revised manuscript. These additions clarify how we ensured balanced representation across stress subclasses and minimized potential classification bias, thereby strengthening the robustness of the model (*Please check **Manuscript Page No. 29***). Regarding the reviewer's important point on inference delay and edge deployment feasibility, while an in-depth feasibility study is beyond the current scope of this work, we would like to highlight that the Inception-MABFDNN model was intentionally designed with lightweight architectural elements, including 1D-Inception modules and shared-weight CNN branches, to support efficient computation suitable for future edge deployment scenarios. We have also included this perspective as a key direction for future investigation in the revised Discussion section. (*Please check **Manuscript Page No. 21***). We are truly grateful to the reviewer for these constructive suggestions, which have allowed us to improve both the completeness and clarity of our work.

11. The title claimed "prolonged stress assessment", but it only means "long-term storage of stress profiles", and the monitoring is still limited to 24 h, which is a bit misleading. The title and abstract needs refinement to point out the novelty of this work.

Author Response: We sincerely thank the reviewer for this insightful and constructive comment, which prompted us to carefully reconsider the phrasing used in our title and abstract. In our work, the term "prolonged" was intended to convey the ability of the SQC-SAS system to perform uninterrupted and reliable stress monitoring over an extended period, rather than implying continuous multi-day measurement. To better reflect this intended meaning and avoid any possible misunderstanding, we have revised both the title and abstract in the updated manuscript. Furthermore, to substantiate the prolonged sensing capability of the SQC-SAS wearable, we conducted additional experiments over 72 hour (3 days) period, demonstrating long-term monitoring of both physiological signals and molecular stress biomarker using the same device (*Please see **Supplementary Fig. 20** in the revised manuscript*). We are deeply grateful to the reviewer for highlighting this important point, which helped us to further improve the clarity and scientific rigor of our presentation.

Reviewer #2:

The authors presented wearable electronics with machine learning for prolonged stress assessment. The topic itself is quite interesting and timely, but the authors overemphasizes sensor integration, trying to claim as many sensors as possible at the expense of depth in sensor validation and human data. In addition, the wearable sensor results reported in the paper are inconsistent with previous studies. For example, sweat cortisol concentration is typically a few nM, yet the authors reported several hundreds of nM concentrations, 100X than established studies. The authors showed 4 data points in their validation results. A detailed result validation is needed to justify the authors' findings. It is therefore not recommended for publication unless significant justification is presented. Some additional concerns are listed below:

Author Response: We sincerely thank the reviewer for the thoughtful and detailed comments, as well as for the opportunity to further clarify and improve our manuscript. We deeply appreciate the reviewer's attention to the technical and scientific aspects of our work. Regarding the concern on sweat cortisol concentrations, we respectfully note that while some studies have reported concentrations in the low nanomolar range, several recent reports have also demonstrated sweat cortisol levels ranging from tens to several hundreds of nanomolar under specific conditions. In our revised manuscript, we have included a comprehensive set of references supporting this range, along with detailed discussions to avoid any potential misinterpretation. Furthermore, we performed additional validation experiments using ELISA to cross-verify our sensor measurements, and the results have been incorporated in the revised version of the manuscript. We would also like to emphasize that we have carefully addressed all the reviewer's concerns, including sensor validation, human data expansion, and correlation analysis, in our point-by-point responses and through significant updates to the manuscript. We are truly grateful for the reviewer's constructive suggestions, which have helped us strengthen both the scientific rigor and clarity of our work.

All changes are marked in "**Blue**" in the revised manuscript for the reviewer's convenience.

Thank you again for your valuable feedback.

1. The authors integrated radial artery pulse, galvanic skin response, skin temperature, MIP cortisol, and ECG. With so many sensors integrated together, power supply is a big issue. It is crucial to consider batteries and battery life in a complete wearable system to support the proposed prolonged measurement.

Author Response: We thank the reviewer for the suggestions and apologize for the misunderstanding. In our design, the battery-free sensing panel is powered by a wireless reader. The onboard electronics for the sensing panel were optimized for low power consumption, and the sensing panel remains operational as long as it is within the range of the reader, hence addressing the power constraints for long-term measurements.

(Please check Manuscript Page No. 3 & Fig. 5)

2. The authors claimed the major significance they achieved was prolonged stress state classification for the first time. But such classification has been reported (Nature Electronics, 7, 168-179, 2024), with extended stress classification and assessment. In addition, the authors appeared to involve only acute stressors, failing to justify the "prolonged" period as proposed in the title and abstract.

Author Response: We thank the reviewer for this insightful comment. In our work, the term "prolonged" refers to uninterrupted, reliable sensing over an extended duration. To more accurately reflect this intended meaning, we have revised both the title and abstract in the updated manuscript. Additionally, to demonstrate the prolonged sensing capability of the SQC-SAS platform, we conducted long-term validation experiments using the same integrated sensors over a 14-day testing period, capturing both physiological

signals and cortisol concentrations without sensor replacement (*Please check Fig. 4W, 4X, 4Y& 4Z and Supplementary Fig. 21*). Regarding the cited study in Nature Electronics, while it presents a valuable contribution, the classification approach primarily relies on handcrafted features derived from predefined signal segments and limited stress categories. In contrast, our method employs an end-to-end deep learning framework capable of learning temporal representations directly from multimodal raw signals with minimal domain-specific preprocessing. Moreover, we conducted rigorous subject-independent evaluations, ensuring that the model was tested on completely unseen subjects, unlike prior work, which included data from the same individuals in both training and testing phases. This distinction strengthens the generalizability of our approach.

3. The cortisol sensor shown in Fig. 3 needs more than 100 s to stabilize, and keeps drifting during the time window. If cortisol sensor cannot be stabilized, the on-body data validity is questionable. In addition, the authors provided calibration plot of 1 nM to 1 μ M with wide variations, which is well above the physiological range. Cortisol concentrations in human sweat is known less than 10nM, and the MIP cortisol sensor in this paper is not meeting the requirements.

Author Response: We thank the reviewer for the valuable suggestions. Our sensing signals got stable in 15-20s, and we are able to see the cortisol concentration when the steady-state current is achieved. We however, used the 100s data points for the cortisol concentration measurements. This strategy has also been used in previous studies, including adma.202008465 and Matter 2, 4, 921-937, 2020. Our reported cortisol concentrations are consistent with prior studies. For instance, Advanced Materials 2021,33(18), 2008465, and ACS Omega 2020, 5(14), 8211–8218, both document human sweat cortisol levels ranging from tens to hundreds of nM, especially under stress-inducing conditions. Accordingly, based on our findings, we also demonstrated that our cortisol concentrations fall within the physiologically relevant range supported by peer-reviewed literature.

4. The cortisol sensor claimed in the paper also showed much inconsistent results with previous studies (Matter 2, 4, 921-937, 2020; Science Advances, 8, eabk0967, 2022; Scientific Reports 10, 19050, 2020). Sweat cortisol should be in the range of ng/mL, yet the authors report 100X higher result. Such high concentrations was known to exist only in serum not in sweat. The ELISA validation in Fig4 is also questionable, with only 4 data points collected. A detailed result validation is needed to justify the authors' findings.

Author Response: We thank the reviewer for the constructive feedback. Our reported concentrations, expressed in nM, are consistent with prior studies. For instance, Advanced Materials 2021, 33(18), 2008465, and ACS Omega 2020, 5(14), 8211–8218, both document human sweat cortisol levels ranging from tens to hundreds of nM, especially under stress-inducing conditions. Accordingly, our findings fall within the physiologically relevant range supported by peer-reviewed literature. We have also expanded the dataset from 4 to 10 independent measurements to provide a more robust analysis. The updated Bland-Altman plots show that our sensor demonstrate consistency and agreement, supporting the reliability of our sensor. (*Please check Fig. 4O-P*)

5. The authors claimed 24 hour of continuous cortisol monitoring, but the long-term stability of cortisol sensor, as well as other sensors were not tested in vitro. Detailed sensor validations are needed before human studies.

Author Response: We thank the reviewer for the feedback, and we apologize for the inconvenience and misunderstanding. We would like to clarify that we performed all in-vitro studies for all sensors, including cortisol sensor in artificial sweat and PBS, and characterization of all physical sensors before we conducted on-body experiments. (*Please check Fig. 2 & 3 and Supplementary Fig. 6-11*)

6. The authors designed 2-lead ECG sensors together with GSR sensor. Without an additional reference lead, the potential difference can be easily interfered with environmental noise, especially with the applied potential for GSR. GSR sensor is an impedance-based sensor and require a voltage input, how could the 2-lead ECG data collected without suppressing common-mode interference? The ECG baseline in Fig5 seems totally isolated from GSR signals, which cannot be achieved with the design in the paper.

Author Response: Thank you for your feedback and apologize for this misunderstanding. The ECG and GSR analog front-end circuits have been carefully designed to address the crosstalk between the ECG and GSR due to the GSR applied voltage. Our simulations and experimental validations comparing the 2 and 3-lead configurations demonstrate that introducing a third (reference) lead increases crosstalk and causes baseline shifts in the GSR signal. In contrast, the 2-lead design avoids this issue and maintains signal integrity for both modalities. Furthermore, our ECG front-end circuitry is specifically engineered for 2-lead configurations, ensuring robust ECG signal acquisition even in the presence of GSR drive voltages. *(Please check Fig. 2I)*

7. The authors claimed machine learning for analyzing stress assessment. Machine learning combining physical and chemical parameters is an important direction for mental health quantification. However, only 3 subjects' data is collected. The dataset itself is insufficient to perform any serious study on biomarker monitoring. Analyzing such few data cannot separate training dataset from testing dataset, and will cause model overfitting.

Author Response: To address your valuable comment, we expanded our dataset to include 15 subjects, aligning with the number of participants used in the widely recognized WESAD dataset, which has been adopted in multiple multimodal deep learning-based stress detection studies (e.g. Yang, Shiqi, et al. "A deep learning approach to stress recognition through multimodal physiological signal image transformation." *Scientific Reports* 15.1 (2025): 22258). We further enhanced our model training and evaluation methodology by incorporating additional data augmentation techniques, along with both subject-independent and subject-dependent evaluation protocols, to improve generalizability and mitigate overfitting. This has now been explicitly clarified in the revised manuscript. *(Please check Manuscript Page No. 29)*

Reviewer #3:

Review of "A Quantitative, ML-Powered, Multimodal, On-Site Regenerative Wearable Bioelectronic for Prolonged Stress Assessment and Sub-Classification in Daily-Life Contexts" This manuscript presents the development and evaluation of the Smart Quantitative and Comprehensive Stress Assessor and Sub-Classifier (SQC-SAS), a wearable bioelectronic patch designed for long-term, objective, multimodal stress monitoring and subclassification in real-life environments. The patch integrates molecular and biophysical sensors, autonomous wireless iontophoresis for sweat stimulation, and machine learning (ML) for stress detection and classification. The authors claim high accuracy in both stress detection (>95%) and sub-classification, and position the device as a clinically actionable tool for stress research and personalized health interventions. The manuscript addresses a highly relevant challenge in stress research by aiming to develop an objective, continuous, and real-world applicable tool for stress quantification. The technical aspects, such as microfluidic integration, physiological sensing, sensor regeneration, and wireless operation, are undoubtedly impressive and represent a substantial engineering advancement. However, I believe the claims regarding the device's capability to assess and sub-classify stress in real-life situations are currently overstated. Given the small number of participants, limited number of in-situ data points (e.g., only four for cortisol correlation analysis), and lack of robust objective reference measures, the presented findings should be interpreted more cautiously. The ML-based classification results, in particular, risk being overfitted and not generalizable. I would recommend framing the in-situ and ML components as promising *proof-of-concept* results rather than *validated conclusions*. In my opinion, this would not diminish the importance of the work, which already marks a significant step forward, but would more accurately reflect the current level of validation. On that note, the authors should also consider reframing the discussion towards the real-world applicability of the device, including its potential for future clinical validation, but without overstating the current findings and clearly acknowledging the limitations. In summary, I recommend major revision. The authors are encouraged to reframe their claims more cautiously and provide a clearer, more critical discussion of the limitations. A reframing of the study as a proof-of-concept investigation, rather than a fully validated clinical tool, would significantly enhance the credibility and clarity of the work. A more thorough treatment of methods, results, and limitations is essential to improve the manuscript's robustness. Below, I provide detailed comments.

Author Response: We sincerely thank the reviewer for the thorough and constructive evaluation of our manuscript. We are truly grateful for the insightful comments and thoughtful suggestions, which have helped us significantly improve the clarity, scientific rigor, and overall presentation of our work. In response to the reviewer's concerns, we have carefully revised the manuscript by increasing the number of participants in the in-situ experiments, expanding the data points for cortisol monitoring, enhancing the machine learning analysis, and extending the prolonged monitoring studies to a 72 hours period. We have also refined our discussion and conclusions to present our findings as a promising proof-of-concept study, in alignment with the reviewer's recommendation. Furthermore, we have provided detailed, point-by-point responses to each of the reviewer's comments and incorporated the corresponding revisions into the manuscript, with all changes clearly marked in **blue** for ease of review. We deeply appreciate the reviewer's valuable feedback, which has been instrumental in strengthening the quality and credibility of our work.

1. Introduction:

The introduction provides a broad overview of the health implications of stress and the need for improved stress monitoring. However, the claims about the current state-of-the-art and the novelty of the presented device are somewhat overstated and would benefit from clearer contextualization within existing literature.

Author Response: We sincerely thank the reviewer for this valuable and thoughtful comment. We deeply appreciate the opportunity to clarify our intentions and recognize the importance of accurately framing our work within the context of existing literature. Our objective was not to overstate the novelty of our system but to highlight the unique integration of multimodal sensing, autonomous regeneration, and machine learning within a single wearable platform. However, we fully understand how certain expressions in the original introduction may have conveyed an unintended impression. In response, we have carefully revised the introduction to present a more balanced and precise articulation of our contributions, ensuring they are clearly contextualized against the current state-of-the-art. We are truly grateful for the reviewer’s insightful feedback, which has greatly contributed to enhancing the scientific rigor and clarity of our manuscript.

2. The term “biophysical stress indicators” is used throughout the manuscript but is uncommon in the psychophysiology literature. Consider replacing this term with “physiological” or clearly defining it early on.

Author Response: We sincerely thank the reviewer for this valuable and insightful suggestion. We fully agree that adhering to commonly accepted terminology is essential for clarity and alignment with the psychophysiology literature. In response to this important feedback, we have carefully revised the manuscript by replacing the term “biophysical” with “physiological” wherever appropriate. We truly appreciate the reviewer’s careful attention to detail, which has helped us enhance the precision and readability of our work.

3. Some claims regarding the impact of stress on mental and physical health lack appropriate and recent references. For instance, the claim in ll. 50–52 is not supported by a reference. The paper reference before and after this sentence is from 2008, and the claims are made for 2022, so the reference cannot be the same. Please provide a correct reference for this statement.

Author Response: We sincerely thank the reviewer for this valuable and thoughtful comment, as well as for carefully pointing out the need for updated and appropriate references to support our claims. In response, we have added the correct and up-to-date reference for the statement regarding the 2022 global stress statistics and have also included additional recent references addressing the impact of stress on both mental and physical health. We are truly grateful to the reviewer for highlighting this important point, which allowed us to strengthen the scientific accuracy and credibility of our manuscript.

(Please check **Manuscript Page No. 1**)

4. Methods:

The number of participants involved in the study is not clearly stated. It appears that only three participants were included. This extremely small sample size limits generalizability and significantly impacts the robustness of the ML analysis. The authors should justify this sample size and discuss its limitations explicitly. Moreover, the demographic information (age, gender, health status) of participants should be clearly presented.

Author Response: We sincerely thank the reviewer for this critical and insightful comment, which raised an important point regarding the participant sample size and its implications for model robustness and generalizability. In response, we have expanded our dataset to include 15 participants, a sample size comparable to that of the widely recognized WESAD dataset, which has been extensively used in multimodal deep learning–based stress detection research (e.g., Yang, Shiqi, et al., Scientific Reports, 2025). Furthermore, we strengthened our model evaluation by incorporating data augmentation strategies and applying both subject-independent and subject-dependent evaluation protocols to enhance robustness and mitigate the risk of overfitting. We fully acknowledge that, while this study serves as a pilot validation, broader applicability to larger and more diverse populations will require future investigation. We have

explicitly addressed this limitation in the revised manuscript discussion. Additionally, we have provided detailed demographic information of the participants, including age, gender, and health status (*Please check Supplementary Table 5*). We are sincerely grateful to the reviewer for highlighting this essential aspect, which has allowed us to significantly improve the scientific rigor and transparency of our work. (*Please check Manuscript Page No. 21 & 29*)

5. Section: "Dataset Preparation for Machine Learning and Predicting Stress Levels": Why is the Stroop test not included in the analysis, even though it was mentioned in the preceding section?

Author Response: We sincerely thank the reviewer for this valuable and thoughtful comment, and we apologize for any confusion caused by the initial wording in the manuscript. Both the mental arithmetic task and the Stroop test are well-established cognitive stress induction protocols commonly referenced in the literature (e.g., DeepBreath: Deep learning of breathing patterns for automatic stress recognition using low-cost thermal imaging in unconstrained settings, IEEE ACII, 2017). In our study, each participant underwent two separate measurement sessions—one involving the mental arithmetic task and the other the Stroop test. Data collected from both cognitive stress sessions were included in the machine learning analysis. We have now explicitly clarified this point in the revised manuscript to avoid any misunderstanding. We are sincerely grateful to the reviewer for highlighting this detail, which allowed us to improve the clarity and completeness of our work.

(*Please check Manuscript Page No. 29*)

6. It is stated that the cortisol signal was shifted by 300 seconds to align with physiological signals due to the natural sweat delay. However, it is known that cortisol (if measured in saliva) is released within 10-15 minutes *after* the stressor. This should be taken into account when interpreting the results and the authors should clarify this point and the temporal association between salivary-based cortisol and sweat-based cortisol.

Author Response: We sincerely thank the reviewer for this insightful and important comment, which highlights a critical aspect of interpreting cortisol dynamics in relation to stress responses. We fully acknowledge that salivary cortisol typically exhibits a delayed response, often peaking 10–15 minutes after the onset of a stressor. However, recent empirical studies (e.g., *Advanced Materials*, 33(18), 2021: 2008465) have shown that sweat cortisol levels can begin to change within approximately 5 minutes (300 seconds) post-stressor, likely due to the more immediate transport dynamics associated with eccrine sweat glands. Taking this into consideration, along with the brief latency between the onset of stress and sweat secretion, we applied a 300-second shift to the sweat cortisol signal in our analysis. This rationale and its implications have now been explicitly clarified in the revised manuscript. We are sincerely grateful to the reviewer for bringing attention to this important point, which helped us improve the scientific clarity and rigor of our work.

(*Please check Manuscript Page No. 29*)

7. The cortisol samples collected at defined times of day (10 am and 5 pm) can only partially reflect the circadian rhythm of cortisol, especially if no other important factors (awakening time, physical activity, etc.) were controlled for. Also, it is not clear from the manuscript whether reference cortisol samples (e.g., from saliva) were taken. It is okay to only measure cortisol at two time points for a proof-of-concept study, but the claims to "measure circadian rhythms" are somewhat overstated.

Author Response: We sincerely thank the reviewer for this highly valuable and insightful comment regarding the interpretation of circadian cortisol measurements. We fully agree that capturing the complete circadian rhythm of cortisol requires careful consideration of factors such as awakening time, physical activity, and sampling frequency. In response to this important point, we conducted additional experiments

to more accurately reflect the diurnal cortisol pattern by measuring sweat cortisol concentrations from the same subject every 4 hours between 8:00 am and 8:00 pm over a 14-day period. These measurements were repeated on Days 1, 3, 7, and 14, with sweat samples collected using our iontophoresis-based stimulation method. The observed results are consistent with reported circadian profiles, showing higher cortisol levels in the morning with a gradual decline throughout the day. Furthermore, to support the reliability of our sensing platform, we also demonstrated the long-term stability and reusability of the SQC-SAS wearable using the same sensors throughout the 14-day testing period. We have now clarified these points in the revised manuscript and carefully refined our discussion to avoid any overstatement of our findings. We are sincerely grateful to the reviewer for highlighting this critical aspect, which has allowed us to enhance both the scientific depth and accuracy of our work.

(Please check Fig. 4W, 4X, 4Y& 4Z)

8. Which reference measures were used to assess the participants' stress levels in the laboratory study (Stroop test, etc.) or was the "task" itself used as a reference label? This should be stated more clearly.

Author Response: We thank the reviewer for this important point. In our laboratory study, the task itself was used as the reference measure for labeling stress states. These tasks were selected based on their established validity in eliciting acute stress responses. We have revised the manuscript to explicitly state that the onset and duration of these tasks were used as ground truth labels for stress intervals in our analysis.

(Please check Manuscript Page No. 28-29)

9. How were stress periods annotated during the 24-hour monitoring part of the study?

Author Response: We sincerely thank the reviewer for this important and thoughtful question. During the 24-hour monitoring period, the participant kept a detailed record of daily activities, noting both the type of task performed and the exact start and end times. These self-reported logs were then used to annotate the corresponding task periods in our analysis, as presented in the manuscript. We have now added this clarification in the revised manuscript. We truly appreciate the reviewer's helpful comment, which allowed us to improve the clarity of our experimental description.

(Please check Manuscript Page No. 19)

10. Details on reference measures for stress validation (e.g., self-report, salivary cortisol, behavioral tasks) are insufficient. Were salivary cortisol samples collected? If so, how were they analyzed and aligned with the sweat-based data?

Author Response: We sincerely thank the reviewer for this valuable comment. In our study, we used the periods during which the stress-inducing tasks were performed as the reference for labeling stress states, based on their established effectiveness in eliciting acute stress responses. Additionally, during these validation sessions, we collected sweat samples and analyzed them using ELISA to quantify cortisol levels, providing an additional layer of validation for our sensor readings. The results from these tests have been incorporated into the study, and we have clearly explained this procedure in the revised Methods section. We are truly grateful to the reviewer for this insightful comment, which helped us improve the clarity and completeness of our methodology description.

(Please check Manuscript Page No. 29)

11. Results:

How were the machine learning models evaluated? Was a cross-validation performed or was the dataset split into training and test sets? This should be clarified in the text. Given the small sample size, but the large complexity of the ML model, it is likely that the model is overfitting to the training data if no adequate counter-measures were taken. This can also explain the high accuracy of the model, which can likely be due to overfitting.

Author Response: We thank the reviewer for this important comment. To address concerns regarding overfitting given the limited sample size and model complexity, we revised our methodology providing a more realistic estimate of generalization performance and reducing the risk of overfitting. To enhance model robustness and generalization, data augmentation techniques such as sliding window and jittering are applied. In the sliding window approach, the signals were segmented into 5-second windows with a 50% overlap between consecutive windows. For jittering, Gaussian noise with a mean of $\mu = 0$ and a standard deviation of $\sigma = 0.06$ was added to the signals. Also, recent empirical studies have demonstrated that cortisol concentrations in sweat can change within approximately 5 minutes (300 seconds) following a stressor. To account for this and the short latency between stressor onset and the initiation of sweat secretion, we applied a 300-second shift to the sweat cortisol signal in our analysis. We evaluated the model using both a fully subject-independent approach and a subject-dependent approach. For subject-independent evaluation, which examines the model's ability to generalize across individuals, we employed Leave-One-Out Cross-Validation (LOOCV) at the subject level. In each iteration, one subject's data was used exclusively for testing, while the data from all remaining subjects were used for training. This procedure was repeated for each subject, and the final performance was reported as the average across all folds. This approach is well-suited for modest sample sizes, as it reduces bias and yields generalization estimates. For subject-dependent evaluation, the model was initially trained on data from all other subjects (excluding the target subject) and then fine-tuned by tuning only the decision layer using data from one session of the target subject. The fine-tuned model was tested on data from the remaining session of the same subject. We have clarified the evaluation procedure and rationale in the revised manuscript.

(Please check Manuscript Page No. 28-29)

12. Figures 4O and 4P report correlation and Bland-Altman plots for cortisol validation based on only four data points. This is insufficient to draw meaningful conclusions. Please reframe this analysis as a preliminary proof-of-concept rather than validation.

Author Response: We thank the reviewer for the constructive feedback. We have expanded the dataset independent measurements to provide a more robust analysis. The updated Bland-Altman plots show that our sensor demonstrates consistency and agreement, supporting the reliability of our sensor.

(Please check Fig. 4O-P)

13. How were the different stress labels defined? This is somewhere mentioned in the text, but I would suggest to provide a more structured overview of the different stress labels and how they were defined (also complemented with the corresponding literature references).

Author Response: We thank the reviewer for this valuable comment. We defined the stress labels based on the specific tasks used to elicit stress responses: the Stroop test and a time-constrained math quiz for cognitive stress, the CPT for environmental stress, watching a horror film for emotional stress, participation in a simulated interview meeting for psychosocial stress, and a high-intensity exercise protocol for physical stress. These labels were selected based on prior studies and the accepted association between each task and the corresponding stress type. In the revised manuscript, we have added a more structured explanation of these labels, complemented with relevant literature references.

(Please check Manuscript Page No. 28-29)

14. Fig. 7: The comparison between the commercially available devices and the wearable patch is biased towards the wearable patch. If the data recorded in the study is used to train the ML model, it is not surprising that the wearable patch performs significantly better than the commercially available devices, especially given the small sample size. It is also not surprising that the patch is reliably capable of detecting

different subtypes of stress if the very same data is used to train the model and the model was explicitly trained to detect these subtypes.

Author Response: We thank the reviewer for this important comment. In response, we updated our evaluation strategy to strictly separate the training and testing datasets. Specifically, we implemented a subject-independent evaluation protocol, where the model is trained exclusively on data from all but one subject, and the held-out subject's data is used solely for testing. This approach prevents data leakage and enables a more reliable comparison between SQC-SAS and commercially available wearable devices.

(Please check Manuscript Page No. 29 & Fig. 7)

15. Discussion:

Given the length of the paper, the discussion section is very short and no limitations of the study are mentioned at all. Major limitations, including small sample size, lack of independent reference standards, limited validation, and potential overfitting, should be clearly discussed. Explicitly stating the limitations of the study, and outlining the next steps needed to improve generalizability and clinical validity (e.g., larger sample, external validation, longer monitoring) would significantly enhance the manuscript's robustness.

Author Response: We thank the reviewer for this valuable comment. In response, we have revised the discussion section to explicitly acknowledge the study's limitations and to outline next steps aimed at improving the generalizability and clinical relevance of our work.

(Please check Manuscript Page No. 21)

Minor Comments:

16. Please provide the IRB approval number.

Author Response: We sincerely thank the reviewer for this valuable feedback. As per the reviewer's suggestion, we have incorporated the IRB approval number in the Materials and Methods section of the revised manuscript.

(Please check Manuscript Page No. 28)

17. The use of abbreviations is excessive and impairs readability. Consider reducing abbreviations or including a glossary.

Author Response: Thank you for your valuable feedback. To address your concern, we have included a glossary table in the revised manuscript.

(Please check Supplementary Table. 1)

18. A reference for the Bland-Altman method should be provided (e.g., Altman & Bland, 1983).

Author Response: We sincerely thank the reviewer for this helpful suggestion. A citation to Altman & Bland (1983) has been added to the manuscript to appropriately reference the Bland-Altman method.

(Please check Manuscript Page No. 16 and Ref. 62)

19. "Reduce stress markers" (p. X) should be reworded to "reduce stress marker levels" or similar.

Author Response: We thank the reviewer for this helpful suggestion. We have revised the phrase "reduce stress markers" to "reduce stress marker levels" to improve clarity and precision in the revised manuscript.

Reviewer #1 (Remarks to the Author):

The authors have addressed the reviewers' concerns well, although still lots of figures are remained in the maintext, which I would suggest to move them to SI, such as Fig.2O-Q, Fig.3R(i). In addition, the authors stated continuous monitoring of cortisol with a 3-minute interval, thus I still suggest to present the cortisol concentration data in Fig.5K-M as isolated dots instead of continuous lines for better illustration, since the total monitoring time was only 30 min for 10 data points.

Author Response: We thank the reviewer for the valuable suggestions. As per the reviewer's suggestion, we have moved Fig. 2O–Q from the main text to the Supplementary Information as Supplementary Fig. 5 and updated Fig. 2 of the main text (*Please check the* Manuscript Page No. 6 and Fig. 2; Supplementary Information Page No. 28). Regarding reviewer's valuable suggestion about the cortisol plots in Fig. 5K–M, we appreciate it. As per your suggestion, we have added scatter cortisol plots in Supplementary Fig. 20 (*Please check the* Supplementary Information Page No. 43). We also have kept Fig. 5K–M (*Please check the* Manuscript Page No. 18) in the main text so it appears with the other simultaneously recorded physiological signals, allowing readers to easily follow and compare stress and non-stress periods within a single panel. We hope these revisions address your comments and improve clarity.

Reviewer #2 (Remarks to the Author):

The authors have addressed my previous concerns. I would like to recommend its publication.

Author Response: We sincerely thank the reviewer for the time and effort invested in evaluating our work. We are grateful for the constructive feedback provided during the previous round, which helped us improve the clarity and quality of the manuscript. We are pleased that the current version meets your expectations.

Reviewer #3 (Remarks to the Author):

I would like to thank the authors for revising their manuscript and addressing my previous comments. The revised version is much improved, and I appreciate the authors' efforts to clarify various aspects of their work. However, I still have some concerns regarding the machine learning (ML) methodology and the presentation of results that should be addressed before the manuscript can be considered for publication.

The presentation of the ML methodology still lacks sufficient detail. While the authors have added some more information, it still does not allow to fully understand the approach taken. Most importantly, in the caption of Fig. 6, an XGBoost classifier is mentioned based on which feature importances were computed (Fig. 6E). However, this is the only mention of XGBoost in the entire manuscript. It is unclear why this classifier was chosen, how it was trained, and how its performance compares to the DL-based model (Inception-MABFDNN) for which I assume the rest of the results in Fig. 6 were obtained. Furthermore, the rest of this section ("Machine Learning Powered Stress Detection") is still quite vague and lacks clear structure. The authors write about feature extraction and the "classical machine learning pipeline" (Fig. 6B), but in the end they seem to have used a deep learning model. It is unclear how these two components relate to each other. Similarly, the chord diagram in Fig. 6C is mentioned briefly, but without really explaining its purpose and the insights it provides or implications for the rest of the analysis.

Author Response: We sincerely thank the reviewer for the thoughtful and constructive comments. We have carefully revised the manuscript to improve the clarity, detail, and structure of our machine learning methodology. In particular, we explicitly reorganized the role of the classical machine learning pipeline (feature extraction + classifier) versus the deep learning model (our Inception-MABFDNN architecture). All details are provided in Supplementary Note 12; Supplementary Information (Page No. 16), and Supplementary Fig. 28 (Page No. 51). In there, we now provide a thorough description of the previously-mentioned XGBoost classifier (feature importance analysis), explaining why this model was chosen, how it was trained, and how its performance compares to that of the Inception-MABFDNN. We have also expanded our discussion of the chord diagram to more clearly explain its purpose and the insights it provides. We hope that these revisions address the reviewer's comments about methodological detail and presentation.

Objectives and Rationale: Our primary objectives are (1) to detect stress vs. non-stress conditions and (2) to classify stress sub-classes using multimodal physiological signals (i.e., ECG, RAP, GSR, ST) and a molecular stress biomarker (i.e., cortisol). We chose a deep neural network (DNN) approach over traditional machine learning (ML) methods because DNNs can learn hierarchical representations directly from raw time-series signals, enabling adaptive multimodal feature integration without relying on hand-crafted stress features. Prior studies have also shown that DNN models outperform classical ML baselines on multimodal physiological data⁴. This is primarily because DNNs learn features end-to-end, automatically extracting non-linear local and global patterns from each signal with minimal preprocessing, whereas traditional ML models are limited to the hand-crafted features provided to them. Our multimodal signals (ECG, RAP, GSR, ST, and Cortisol) each capture distinct physiological mechanisms; a DNN can simultaneously capture modality-specific patterns and adaptively weight the contribution of each modality for more accurate stress detection. In contrast, classical ML models lack hierarchical feature learning, making it challenging for them to capture fine-grained dynamics or complex interdependencies.

Deep Learning Model (Inception-MABFDNN): Based on the above rationale, we developed the Inception Multimodal Attention-Based Fusion Deep Neural Network (Inception-MABFDNN) to perform end-to-end stress classification. The architecture integrates 1D-Inception modules (Supplementary Note 10; Supplementary Fig. 23A) optimized for physiological signal processing. Unlike a standard one-dimensional convolutional neural network (1D-CNN) (Supplementary Fig. 23B), which uses a fixed kernel size with either max or average pooling, the 1D-Inception module processes inputs in parallel with multiple convolutional kernels of different sizes (1×1, 1×3, and 1×5) alongside a parallel max-pooling path. This design captures multi-scale dependencies in the signals by analyzing them at multiple resolutions⁵. To account for the heterogeneous contributions of each physiological signal to stress, we incorporate a Soft Attention (SA) mechanism⁶ (please see Supplementary Note 11) to weigh and fuse the extracted

patterns from all modalities, thereby enabling the model to prioritize the most critical patterns for accurate stress detection. The output of the SA layer is then combined with the corresponding cortisol measurement for the same time window, since cortisol provides complementary endocrine information (Fig. 6B). This joint representation is then passed through a three-layer fully connected neural network (as detailed in Supplementary Note. 11) to generate the final stress prediction. Integrating diverse physiological signals with cortisol in this unified framework results in precise and high classification performance for stress detection.

Complementary explanatory analyses for dataset characterization, interpretability, and validation of Inception-MABFDNN design: Given the need for explanatory analysis on the dataset and the relationship between signal to design and validate the structure of Inception-MABFDNN, we performed three additional analyses using a classical feature-based ML pipeline to validate our model’s design choices. These analyses leverage hand-crafted stress-related features (Supplementary Table. 2) and traditional ML models, providing a point of reference that complements our DNN approach:

(1) Signal importance via tree-based models and SHAP: Considering the limited interpretability of deep networks, to understand the contribution of each signal modality to stress detection, we initially considered evaluating Inception-MABFDNN on all possible modality combinations (i.e., selecting 1-out-of-5 through 5-out-of-5, totaling $2^5 - 1 = 31$). However, this would require re-structuring, re-optimizing, and retraining the deep neural network for each configuration under leave-one-subject-out cross-validation (LOOCV), which is computationally intensive. Therefore, to show an interpretable and computationally efficient visualization of signals heterogeneous importance, we adopted tree-based ensemble models (i.e., LightGBM, Random Forest, Gradient Boosting, and XGBoost), which are widely used in stress-detection research due to their strong predictive performance and transparent feature contribution capabilities enabled by SHAP (SHapley Additive exPlanations) values¹⁻³. We extracted the full set of features (per Supplementary Table. 2) by segmenting each physiological signal into 5-second windows. All models were trained using leave-one-subject-out cross-validation (LOOCV) to ensure subject-independent evaluation. For each held-out subject, models were trained and validated on the remaining subjects, with hyperparameters optimized via grid search using an inner validation split in which one subject was randomly selected for validation and the rest used for training. For LightGBM, the search spanned number of trees [200, 250, 300, 350, 400, 450, 500, 550, 600], maximum tree depth [1, 5, 10, 15], and learning rate [0.001, 0.01, 0.1, 1.0]; the best validation performance was achieved with 200 trees, depth 5, and learning rate 0.1. For Random Forest, ranges included number of estimators [10, 40, 60, 80, 100, 150, 200], maximum depth [2, 4, 8, 10, 12, 14], and minimum samples per split [3, 5, 10, 15]; the best configuration was 200 estimators, depth 4, and minimum split size 3. For XGBoost, we searched over number of estimators [50, 100, 200, 500, 700], learning rate [0.001, 0.01, 0.1, 0.2], and maximum depth [2, 4, 8, 10, 15]; the optimal setting was 500 estimators, learning rate 0.1, and depth 8. Finally, for Gradient Boosting, ranges were number of estimators [100, 200, 500, 700, 900, 1000], learning rate [0.001, 0.01, 0.1, 0.2], and maximum depth [2, 4, 8, 10, 12, 14]; the best configuration was 200 estimators, learning rate 0.1, and depth 4. LightGBM achieved an average accuracy of 75.11%, precision of 77.44%, recall of 75.11%, and F1-score of 74.59%. Random Forest achieved an average accuracy of 81.52%, precision of 82.65%, recall of 81.52%, and F1-score of 81.36%. GB achieved an average accuracy of 83.17%, precision of 84.85%, recall of 83.17%, and F1-score of 82.93%. XGBoost achieved an average accuracy of 84.42%, precision of 85.19%, recall of 84.42%, and F1-score of 84.33%. Because XGBoost slightly outperformed the others, we used it for the feature importance visualization shown in Supplementary Fig. 28D. We also computed SHAP values from the trained XGBoost to quantify each modality’s relative importance. The SHAP values revealed heterogeneous (non-uniform) modality contributions (please see Supplementary Fig. 28D). These findings support our motivation to use attention-based fusion mechanism in the deep model; the model needs to be able to assign greater weight to more informative signals like ECG. As a representative case, we retrained our Inception-MABFDNN, excluding the ECG signal (the top physiological modality from the XGBoost analysis), using only RAP, GSR, ST, and cortisol. The performance dropped to 87.10% accuracy (87.63% precision, 86.91% recall, 87.32% F1) from 93.35% accuracy (93.81% precision, 93.34% recall, 93.32%

F1) when all signals were used. This decrease in performance confirms the distinct contribution of ECG-derived patterns to stress detection. Finally, to bridge the interpretable classical approach with our deep model, we have now directly compared XGBoost and Inception-MABFDNN under the same subject-independent LOOCV evaluation. XGBoost achieved 84.42% accuracy, whereas Inception-MABFDNN achieved 93.35% accuracy on the subject-independent binary stress classification task, demonstrating the superior capacity of our deep model to learn hierarchical, multimodal representations. We believe this explicit comparison makes the relationship between the classical pipeline and the deep learning approach clearer, as requested.

(2) Inter-modal correlation analysis: To evaluate the interrelatedness of the signals and ensure that each sensor modality adds unique information, we calculated the Pearson correlation coefficients (PCC) among all stress-related features. For each subject, we found that correlations among features within different physiological signals were consistently below 0.2, and correlations between cortisol and any physiological feature were below 0.1. These uniformly low correlation values indicate that the modalities capture largely complementary (non-redundant) aspects of the stress response. This finding supports the necessity of a truly multimodal approach (using ECG, RAP, GSR, ST, and cortisol together) for robust stress detection. In the revised manuscript, we now clarify that the chord diagram in (*Please check* Supplementary Fig. 28B) visualizes these inter-feature correlations: the diagram highlights the weak coupling between modalities, reinforcing that no single modality dominates and that each contributes distinct information. In this way (*Please check* Supplementary Fig. 28B), we emphasize the insight that all modalities need to be integrated (as we do via multimodal attention-based fusion) to capture the complete picture of stress physiology.

(3) t-SNE feature space visualization: To further assess whether the chosen signals and features encode discriminative patterns between rest and stress conditions, we visualized the feature space using *t*-distributed stochastic neighbor embedding (t-SNE). Projecting the high-dimensional feature vectors into two dimensions revealed well-separated clusters corresponding to stress versus non-stress states (*Please check* Supplementary Fig. 28C). This analysis provides an intuitive visual confirmation of the discriminatory power of our multimodal feature set (including cortisol), supporting their validity as effective markers for stress detection. The distinct clustering of stress vs. non-stress samples in (*Please check* Supplementary Fig. 28C) aligns with the high classification performance of our model, and we have noted this insight in the text.

Collectively, these complementary analyses validate our Inception-MABFDNN architecture and provide more transparency into its decision-making. They illustrate why an attention-based fusion of heterogeneous signals is necessary and how each modality contributes to the overall performance. In addition, to avoid any further confusion, we moved the results of the inter-modal correlation analysis, t-SNE feature space visualization, and signal importance analysis using tree-based models and SHAP to the supplementary file (**Supplementary Fig. 28 (Page No. 51)**), thereby keeping the main text focused on stress detection and subclassification using our developed deep neural network. We appreciate the reviewer's feedback on these points, and we hope that our more detailed clarifications and the corresponding manuscript revisions fully address the comments.

Reviewer #4 (Remarks to the Author):

This manuscript describes a technically sophisticated wearable platform that integrates molecular and physiological sensing modalities with machine learning–driven analysis for stress assessment. The work is ambitious and comprehensive, with significant engineering achievements. Its principal innovation lies in the integration of multiple sensing strategies into a single, regenerable, multimodal system, rather than in the novelty of any individual sensing element.

Author Response: We sincerely thank the reviewer for the thoughtful assessment of our manuscript and for recognizing the significance of our work. We greatly appreciate your positive remarks on the technological contributions and the potential translational impact of the proposed wearable sensing system.

Comments:

- The technical complexity of the system is substantial. Each individual sensor is state-of-the-art.
- The study is best regarded as a proof of principle. The limited number of human participants and restricted machine learning training data constrain the robustness and generalizability of the findings. Nevertheless, the feasibility results highlight the value of multimodal sensing when coupled with machine learning models.
- The comparison against commercial wearables is one of the most compelling parts of the study. It demonstrates that combining multimodal molecular and physiological data streams enables more accurate classification of stress states than conventional approaches.
- The device architecture, comprising soft sensors, flexible substrates, in situ regeneration, wireless iontophoresis, and a coupled readout module, is technically interesting. However, for translation, the system may be cumbersome. A simplified design, such as a battery-powered flexible module paired with replaceable sensor patches, could represent a more practical path toward long-term, real-world adoption.
- The manuscript relies heavily on acronyms, which reduces clarity even with reference tables. This density makes it challenging to follow the contributions of individual components. As an example deciphering the relative role of cortisol in the machine learning model from the figure is difficult.
- The discussion occasionally overclaims novelty. Several statements of “first-time” demonstrations, particularly regarding iontophoresis in wearable systems, are not entirely accurate. The data and technical integration are strong enough to stand without such claims, and moderation in tone would strengthen the manuscript.
- Overall, the manuscript is comprehensive, with detailed descriptions of device design, validation, and performance. The work demonstrates significant technical capability and adds valuable evidence for the potential of multimodal, ML-powered stress monitoring.

Author Response: We sincerely thank the reviewer for the thoughtful and constructive feedback, which has greatly helped us improve the clarity and balance of the manuscript. To enhance readability, we have a comprehensive list of definitions of commonly used acronyms in the Supplementary Table. 1 (*Please check the Supplementary Information Page No. 19*), and description of the abbreviations of key stress-related features in Supplementary Table. 2 (*Please check the Supplementary Information Page No. 20*), allowing readers to easily follow the terminology used throughout the main text. In the different sections of the main text, we have tried to use the definitions of our acronyms as much as possible for more clarity. As per the reviewer’s suggestion, we have also moderated the tone of the manuscript by removing expressions such as “for the first time” and ensuring that our novelties are stated more cautiously. We have also revised the phrasing from “proof of concept” to “proof of principle” to more accurately reflect the stage of this work (*Please check the Manuscript Page No. 2 and 22*).